

# The influence of Holocene vegetation changes on topography
# and erosion rates: A case study at Walnut Gulch Experimental
# Watershed, Arizona
**Jon D. Pelletier[1*], Mary H. Nichols[2], Mark A. Nearing[2]**
[1]*Department of Geosciences, University of Arizona, Gould-Simpson Building, 1040 East Fourth*
*Street, Tucson, Arizona 85721-0077, USA*
[2]*USDA Southwest Watershed Research Center, 2000 East Allen Rd., Tucson, Arizona 85719,*
*USA*
*corresponding author;* email: jdpellet@email.arizona.edu
**Abstract**
Quantifying how landscapes have responded and will respond to vegetation changes is an
essential goal of geomorphology. The Walnut Gulch Experimental Watershed offers a unique
opportunity to quantify the impact of vegetation changes on landscape evolution over geologic
time scales. The Walnut Gulch Experimental Watershed (WGEW) is dominated by grasslands at
high elevations and shrublands at low elevations. Paleovegetation data suggest that portions of
WGEW higher than approximately 1430 m a.s.l. have been grasslands and/or woodlands
throughout the late Quaternary, while elevations lower than 1430 m a.s.l. changed from a
grassland/woodland to a shrubland c. 2-4 ka. Elevations below 1430 m a.s.l. have decadal time-
scale erosion rates approximately ten times higher, drainage densities approximately three times
higher, and hillslope-scale relief approximately three times lower than elevations above 1430 m.
We leverage the abundant geomorphic data collected at WGEW over the past several decades to



calibrate a mathematical model that predicts the equilibrium drainage density in shrublands and
grasslands/woodlands at WGEW. We use this model to test the hypothesis that the difference in
drainage density between the shrublands and grassland/woodlands at WGEW is partly the result
of a late Holocene vegetation change in the lower elevations of WGEW, using the upper
elevations as a control. Model predictions for the increase in drainage density associated with the
shift from grasslands/woodlands to shrublands are consistent with measured values. Using
modern erosion rates and the magnitude of relief reduction associated with the transition from
grasslands/woodlands to shrublands, we estimate the timing of the grassland-to-shrubland
transition in the lower elevations of WGEW to be approximately 3 ka, i.e., broadly consistent
with paleovegetation studies. Our results provide support for the hypothesis that common
vegetation changes in semi-arid environments (e.g. from grassland to shrubland) can change
erosion rates by more than an order of magnitude, with important consequences for landscape
morphology.
*Keywords: landscape evolution, drainage density, vegetation cover, Walnut Gulch Experimental*
*Watershed*
**1. Introduction**
**1.1. Problem statement**

Understanding how climate change controls landscape evolution is a central problem in

geomorphology. Climate changes are multifaceted, with changes in temperature (mean and
variability), precipitation (mean and variability) and vegetation cover (type and density) often
occurring simultaneously. The multifaceted nature of climatic changes can make it difficult to
identify which aspects of climate change are most important in driving landscape modification in



specific cases. Yet, given the accelerated climatic changes expected to occur in the coming
decades, understanding how landforms are likely to respond to specific climatic drivers, acting
alone or in concert, is critically important to society (e.g., Pelletier et al., 2015).

In the southwestern U.S. the existence of an extensive, regionally correlative fan and

valley floor depositional unit (the Q3a unit of Bull, 1991) suggests that the Pleistocene-to-
Holocene transition was a major perturbation in what are now semi-arid shrubland-dominated
landscapes but which were predominantly pinyon-juniper woodlands at the Last Glacial
Maximum (LGM). Where rates of aggradation in modern shrubland drainage basins have been
measured, the Pleistocene-to-Holocene transition was associated with more than an order-of-
magnitude increase above either LGM or mid-to-late Holocene rates (e.g., Fig. 3.11 of Weldon,
1986). The retreat of grasslands and woodlands to higher elevations and their replacement by
shrublands likely exposed elevations of the southwestern U.S. between approximately 800 m and
at least 1400 m a.s.l. to significant increases in percent bare ground, thus modifying the rainfall-
runoff partitioning of hillslopes and their resistance to fluvial/slope-wash erosion. In especially
arid areas of the southwestern U.S. such as the central Mojave Desert, the range of elevations
affected by late Quaternary conversions of grasslands/woodlands to shrublands extends to
elevations as high as 1800 m a.s.l. (Pelletier, 2014).

Given the strong correlation between percent bare ground and drainage density in the

southwestern U.S. (Melton, 1957), it has been hypothesized that modern shrublands sufficiently
high in elevation to have been grasslands or woodlands at LGM underwent large increases in
drainage density during the Pleistocene-to-Holocene transition. Such an expansion of the fluvial
drainage network could have mobilized hillslope deposits stored as colluvium during the last
glacial epoch, mobilizing large volumes of sediment into the fluvial system during the transition





to the present interglacial (Bull, 1991; Pelletier, 2014). This hypothesis is consistent with
measured ages of the onset of aggradation in valley floor and alluvial fan depositional zones in
the central Mojave Desert, in which aggradation occurs earliest (c. 15 ka) in depositional zones
fed by source regions with relatively low elevations in the 800-1800 m a.s.l. range and
progressively later in areas fed by eroding regions at higher elevations (Pelletier, 2014).
Alternative explanations for the punctuated nature of late Quaternary aggradation in the
southwestern U.S. invoke changes in the frequency and/or magnitude of floods (Antinao and
McDonald, 2013b) and argue that hillslope vegetation changes played a limited role (Antinao
and McDonald, 2013a). To date, tests of the paleovegetation change hypothesis in the
southwestern U.S. have been limited to studies of the timing of deposition on valley-floor
channels and alluvial fans. Erosion of the source drainage basins themselves has been relatively
understudied.

The Walnut Gulch Experimental Watershed (WGEW) provides an excellent opportunity

to test the paleovegetation change hypothesis in a drainage basin that has been extensively
monitored for decades. The western, low-elevation portion of WGEW is currently a shrubland
while the eastern portion is predominantly a grassland (a small area of woodland occupies the
highest elevations). Paleovegetation data, however, indicate that all of WGEW was a grassland
or woodland until just a few thousand years ago. Scientists working at WGEW have measured
rates of sediment export from watersheds using sediment samplers (e.g., Nichols et al., 2008) and
rates of sediment redistribution within watersheds using anthropogenic radionuclides (e.g., $^{137}$Cs)
and rare-earth element tracers (e.g., Nearing et al., 2005;. Polyakov et al., 2009). These data, in
addition to the results of analyses of airborne lidar data presented here, make it possible to
calibrate every parameter of a mathematical model that predicts the equilibrium drainage density



of the landscape under different dominant vegetation types. In this paper we use this
mathematical model to test the hypothesis that grassland/woodland-to-shrubland vegetation
changes in the lower elevations of WGEW drove large increases in drainage density and erosion
rates and a decrease in hillslope-scale relief.
**1.2. Study Site**
*1.2.1. Geology and Soils*

The Walnut Gulch Experimental Watershed (WGEW) is part of the U.S. Department of

Agriculture (USDA) Agricultural Research Service's (ARS) Southwest Watershed Research
Center (SWRC). WGEW is located at the boundary between the Chihuahuan and Sonoran
Deserts and elevations of between approximately 1300 and 1600 m a.s.l. The approximately 150
km$^2$ watershed has a planar geometry at large spatial scales, dipping to the WSW with a slope of
approximately 1.5%, i.e., approximately 230 m of elevation change over a distance of 15 km.

The bedrock geology of WGEW includes sedimentary, plutonic, and volcanic rocks of

Precambrian to late Cenozoic age (Fig. 1). Due to the complex nature of the rock types exposed
in the southern portion of the watershed, we focus this study on the northern portion of the
watershed, which is dominated by the Gleeson Road Conglomerate (GRC).

The GRC was derived primarily from erosion of the Dragoon Mountains to the east of

WGEW and is estimated to be Plio-Pleistocene in age by Osterkamp (2008), who noted: "The
upper part of the Gleeson Road Conglomerate is probably equivalent both stratigraphically and
in age to the Plio-Pleistocene upper basin fill of Brown et al. (1966). To the west and northwest,
along the axis of the San Pedro River Valley, the upper part of the Gleeson Road Conglomerate
grades into fine-grained fluviatile and lacustrine beds of the Plio-Pleistocene St. David
Formation (Gray 1965, Melton 1965)." The GRC dips gently (5-8°) to the north and northwest



(Gilluly, 1956, plate 5), a fact which may contribute to the generally longer and more gently
sloping south-facing hillslopes relative to north-facing hillslopes in the watershed. The tilted
strata of the GRC were beveled to a gently sloping topographic surface and incised into during
Quaternary time. Incision of the GRC was driven by uplift/tilting of the fan and/or by incision of
the San Pedro River, triggering headward erosion of Walnut Gulch and its tributaries.
The northern portion of WGEW is composed of the Whetstone Pediment of Bryan (1926)
and is divided into two parts: the "dissected Whetstone Pediment" (DWP) at elevations below
approximately 1430 m a.s.l. and the "upper Whetstone Pediment" (UWP) at higher elevations
(Fig. 1). The DWP is distinguished from the UWP by its lower relief and less well-developed
soils. Differences between the DWP and UWP have been attributed primarily to headward
extension of tributaries resulting from late Quaternary incision of the San Pedro River (Cooley,
1968) as well as renewed river and tributary incision following livestock grazing beginning c.
1880 (Renard et al., 1993; Osterkamp, 2008). However, the boundary between the DWP and
UWP coincides with the transition from the modern shrubland to the grassland (Fig. 2). As such,
we hypothesize that vegetation cover and its changes over geologic time scales have also
contributed to differences between the DWP and UWP.
Deep sandy gravel loams of the Blacktail-Elgin-Stronghold-McAllister-Bernardino
Group occur in areas of the UWP. In the lower, western part of the watershed, soils are in the
Luckyhills-McNeal Group. Soils of the Luckyhills-McNeal Group tend to be sandy and gravelly
loams that are immature compared with soils of the Blacktail-Elgin-Stronghold-McAllister-
Bernardino Group. The A horizon of the Luckyhills-McNeal Group is typically absent, having
been removed by late Quaternary erosion (Breckenfeld, 1994). The boundary between these two



soil groups coincides with the boundary between the DWP and UWP and the transition from the
modern shrubland to the grassland.
*1.2.3. Climate and Vegetation*
Mean annual temperature at Tombstone (located in the western portion of WGEW at an
elevation of 1384 m a.s.l.) is 17.6°C and mean annual precipitation is approximately 300 mm.
There is both a winter and summer rainy season, but approximately 70% of rainfall occurs in the
summer months and the greater intensity of summer storms means that runoff results almost
exclusively during the summer season of July through September. Mean annual precipitation is
approximately 10% higher at the highest elevations of the watershed relative to the lowest
elevations (Nearing et al., 2015).
Modern vegetation cover in the U.S.-Mexico Borderlands region closely follows
elevation, with desert scrub occurring below approximately 1500 m, grasslands from 1400–
1700 m, lower encinal ("encina" is Spanish for "oak") from 1700–2600 m, upper encinal from
1900-2600 m, and forest from 2200-2600 m (Wagner, 1977). For each zone, the highest
elevations are reached on dry, south- and west-facing slopes and the lowest elevations on north-
facing slopes and valley bottoms.
Paleovegetation records from the Borderlands region suggest that the western portions of
WGEW have transitioned from a grasslands/woodland to a shrubland over the past few thousand
years, while the eastern half of the watershed has been a grassland/woodland for at least the past
40,000 yr and likely since the penultimate interglacial period. Low stalagmite $^{18}$O values at the
Last Glacial Maximum (LGM) in the Cave of the Bells paleoclimate record indicate that
conditions were much wetter and cooler during LGM (Wagner et al., 2010), in agreement with
paleovegetation studies (Betancourt et al., 1990; Anderson, 1993; Betancourt et al., 2001;



Arundel, 2002; Holmgren et al., 2003; Holmgren, 2005). Packrat midden records indicate the
presence of grasslands and/or pinyon–juniper woodlands at LGM in what is currently
Chihuahuan  desert  scrub at elevations of 1200-1400 m a.s.l. (Betancourt  et al., 2001).
Holmgren (2005) documented the presence of the primary grass species at WGEW (*Bouteloua*
*eriopoda*) (currently abundant only at elevations above approximately 1430 m a.s.l.) at an
elevation of 1287 m c. 4750 $^{14}$C yr B.P. Elements of the modern shrubland, such as *Larrea*
*tridentate*, appeared as late as 2190 $^{14}$C yr B.P. at 1287 m a.s.l. based on macrofossils, but may have
been present as early as 4095 $^{14}$C yr B.P. based on pollen. As such, the latest Quaternary transition
from grasslands/woodlands to shrublands in WGEW occurred gradually and was completed only
within the past few thousand years.
***1.2.4. Intensive monitoring sites***

WGEW is home to two intensive monitoring sites, one in the shrubland of the DWP and

the other in the grassland of the UWP. These sites provide the data necessary, in conjunction
with the topographic analyses presented here, to calibrate a mathematical model that predicts the
equilibrium drainage density as a function of vegetation cover and to test the hypothesis that
differences in landscape morphology and erosion rates between the northwestern and
northeastern portions of WGEW are partly the result of a transition from grassland/woodland to
shrubland within the past few thousand years in the northwestern portion of the drainage basin
that did not occur in the northeastern portion.

Watersheds 63.102, 63.103, 63.104, 63.105, and 63.106 are located in shrublands at an

elevation of approximately 1370 m a.s.l. in what is referred to as "Lucky Hills," which has been
the site of a variety of intensive scientific studies since the 1960s. Cover during the rainy season
at Lucky Hills is approximately 25% bare soil, 25% canopy, and 50% erosion pavement (rocks).

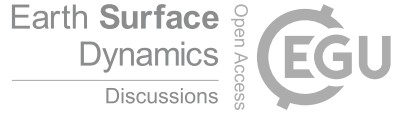

Dominant vegetation includes: Creosote (*Larrea tridentata*, shrub) and Whitethorn (*Acacia*
*constricta*, shrub), with lesser populations of Desert Zinnia (*Zinnia acerosa*, shrub), Tarbush
(*Flourensia cernua*, shrub), and sparse Black Grama (*Bouteloua eriopoda*, grass). The matrix
material of surface layer is composed of 60% sand, 25% silt, and 15% clay. Sediment from the
watershed is monitored with a supercritical flume with an automatic traversing slot sampler
(Simanton et al., 1993).
Watershed 63.112 is located in the Kendall grassland site at WGEW, approximately 10
km east of Lucky Hills and at an elevation of approximately 1525 m. The site is predominantly
covered by grass and forbs with some shrubs and succulents with a combined canopy cover of
approximately 35%. Ground cover during the rainy season has been measured at 28% rock, 42%
litter, and 14% plant basal cover (Nearing et al., 2007). Compared to the 25% bare soil at Lucky
Hills, the bare soil exposed at Kendall is negligible (i.e. a few percent or less). Historically, the
dominant desert grassland bunchgrasses at the site have been black grama (*Bouteloua eriopoda*),
side-oats grama (*B. curtipendula*), three-awn (*Aristida sp.*), and cane beardgrass (*Bothriochloa*
*barbinodis*) (King et al., 2008), and more recently, Lehmann lovegrass (*Eragrostis lehmanniana*)
(Moran et al., 2009).
Nearing et al. (2005) used spatially distributed [137]Cs measurements to quantify
fluvial/slope-wash erosion and deposition rates within and from watersheds 63.103 (Lucky Hills)
and 63.112 (Kendall). Nearing et al. (2005) found that in Lucky Hills the fraction of the drainage
basin experiencing erosion was much higher (85%) than in Kendall (53%), where erosion and
deposition rates were lower and approximately balanced such that the rate of net sediment
exported from the Kendall watershed was more than a factor of ten lower than from the Lucky
Hills watershed. These observations are consistent with sediment yields measured from 1995-



2005 that are more than ten times higher in drainage basins of similar size at Lucky Hills (231 t
km$^{-2}$ yr$^{-1}$ in  watershed 102 (area of 1.46 Ha)) than at Kendall (7 t km$^{-2}$ yr$^{-1}$ in watershed 112
(area of 1.86 Ha)) (Nearing et al., 2007). Assuming a soil bulk density of 1500 kg m$^{-3}$, these
sediment yields correspond to mean erosion rates of approximately $1.5 \times 10^{-4}$ m yr$^{-1}$ in the
shrublands of Lucky Hills and $5 \times 10^{-6}$ m yr$^{-1}$ in the grasslands of Kendall (Table 1). Hillslopes are
slightly steeper in Kendall, so if anything we would expect erosion rates to be higher at Kendall
than at Lucky Hills if slope gradient were the dominant factor in controlling erosion rates.
Nearing et al. (2005) interpreted the differences in erosion rates between Lucky Hills and
Kendall to be primarily a function of vegetation cover, i.e. "hydrologic response differences as a
function of vegetation differences are probably largely responsible for the differences in hillslope
erosion rates between the two watersheds. If flows are more concentrated and vegetative cover is
less, as on the Lucky Hills site, flow shear stresses and stream power will tend to be greater,
resulting in a greater hydrologic potential for erosion. Also important is probably the higher litter
cover and plant basal area cover on the grassland site that would have a direct protective effect
against erosion." This interpretation is consistent with the conclusions of Abrahams et al. (1995)
and Parsons et al. (1996), who emphasized the role of vegetation cover in controlling
fluvial/slope-wash erosion rates in their plot-scale studies at WGEW. In this paper we explore
the implications of these erosion rate differences for landscape morphology and topographic
evolution of WGEW over geologic time scales.

**2. Methods**
**2.1. Topographic analysis**





In this section we describe the methods used to quantify the similarities and differences in

landscape morphology between the modern grassland and shrubland sites, with an eye toward

providing the data necessary to calibrate the mathematical model described in section 2.2. In all

of the topographic analyses described in this section we used a 1 m pixel$^{-1}$ Digital Elevation

Model (DEM) for WGEW derived from airborne laser swath mapping (Heilman et al., 2008).

Prior to the analyses, the DEM was smoothed with an Optimal Weiner Filter (OWF), following

the approach of Pelletier (2013), to remove small-scale variability related to errors related to

distinguishing ground from vegetation points and other imperfections of the lidar-derived DEM.

The smoothing did not significantly alter slope gradients but did significantly reduce anomalous

curvature values related to DEM imperfections.

The drainage density in the shrubland areas appears to be much higher than in the

grassland areas (Fig. 3). To quantify this difference we used the method developed by Pelletier

(2013) to identify the network of valley bottoms (i.e. where water flow is localized and fluvial

processes are responsible for the majority of erosion) in the vicinity of Lucky Hills and Kendall.

In this method, the DEM is filtered using the OWF, the contour curvature is computed at every

pixel, and valley heads are identified as the areas closest to the divides where the contour

curvature exceeds a user-defined threshold value. In Pelletier (2013) and in this paper a threshold

contour curvature of 0.1 m$^{-1}$ was used for valley head identification (i.e., the method of Pelletier

(2013) was used without modification or parameter tuning). Once valley heads are identified, a

multiple-flow-direction routing algorithm is used to route a unit of runoff from each valley head

to identify the valley bottoms downstream. In this study we used the distance along flow lines

from topographic divides to the first valley bottom as a measure of the extent of the drainage

network. We computed the mean value of this hillslope length for all pixels entering valley



heads. This mean value can be compared to the prediction of a mathematical model that
computes the mean distance along flow lines from topographic divides to valley heads as a
function of colluvial and fluvial transport coeffiicients. Hillslope length measured in this way is
inversely related to drainage density (Horton, 1945). Its mean value contains information
equivalent to drainage density, but it has the advantage that it is a mappable quantity (Tucker et
al., 2001). The drainage network analysis was performed on representative 1.6 x 1.6 km areas in
the vicinity of Lucky Hills and Kendall to quantify the difference in drainage density between
shrubland and grassland areas within WGEW.
Relief was mapped as the difference between the highest elevation upstream from each
pixel along flow lines. The results of the drainage network identification procedure described
above were used to limit this analysis to the hillslope pixels only. We then computed the mean
hillslope relief within 10 m elevation bins from 1320 m to 1550 m a.s.l. The resulting graph
quantifies how the mean hillslope relief varies with elevation across the shrubland-to-grassland
transition.
We also computed the mean value of the along-channel slope gradient and curvature (i.e.
the Laplacian) as functions of contributing area. Differences in mean curvature as a function of
contributing area provide a quantitative signature of how late Holocene vegetation changes have
modified the landscape morphology in the vicinity of the hillslope-to-valley-bottom transition.
**2.2. Mathematical modeling**
In this section we describe the mathematical model used to quantify erosion over
geologic time scales and its dependence on landscape morphology at WGEW. The mathematical
model is used to predict the equilibrium drainage density, quantified as the mean distance along
flow lines from divides to valley bottoms, in both shrublands and grasslands, in order to test the



hypothesis that the difference in drainage density observed between the shrublands and
grasslands at WGEW can be attributed, in part, to late Holocene vegetation changes in the
shrubland portion of the watershed.

Erosion at WGEW over geologic time scales can be approximated as the sum of erosion

due to colluvial and fluvial/slope-wash processes. Sediment transport by colluvial processes
leads to a diffusion equation for topography if slope are uniformly soil-mantled and topographic
gradients are modest (Culling, 1960):

$E_c = -D\nabla^2 z$                                        (1)

where $E_c$ is the erosion rate by colluvial processes (defined to be positive if the landscape is
eroding), $D$ is diffusivity in m² yr⁻¹, and $z$ is elevation in m. Equation (1) assumes that colluvial
sediment flux is equal to the product of a coefficient (i.e., the diffusivity $D$) and the local slope
gradient. This assumption is reasonable for WGEW, where hillslopes are uniformly soil mantled
and the mean hillslope gradient is 7%.

We assume that fluvial/slope-wash processes in WGEW can be approximated as

transport limited. We use the term fluvial/slope-wash to refer to all sediment transport by
flowing water, wherever it occurs along the continuum from hillslopes (i.e., as sheet and rill
flow) to channels (i.e., fluvial erosion). A transport-limited condition applies to landscapes in
which most of the sediment is transported as bed-material load and sediment is readily deposited
if the shear stress by flowing water declines with increasing distance along flow pathways. In the
alternative detachment-limited end-member model of fluvial/slope-wash erosion, the shear stress
required to detach sediment is much larger than the shear stress required to transport it, hence
sediment redeposition is rare or nonexistent once detachment/entrainment occurs. Pelletier
(2012) addressed the geomorphic conditions under which transport-limited versus detachment-





limited conditions are likely to occur, taking into account data for the relative proportion of
sediment transported as bed-material load versus wash load, among other factors, using WGEW
as a case study. He concluded that among these two end-member models, WGEW is most
accurately considered to be transport limited.
The assumption of transport-limited conditions implies that fluvial/slope-wash erosion,
$E_f$, is equal to the divergence of the fluvial/slope-wash volumetric unit sediment flux, $\mathbf{q}_s$ (the
volumetric sediment flux per unit width of water flow):
$$E_f = \frac{1}{\varepsilon_0} \nabla \cdot \mathbf{q}_s .$$   (2)
where $\varepsilon_0$ is the grain packing density (assumed here to be 0.55, e.g., equivalent to a bulk density
of 1500 kg m$^{-3}$ for a grain density of 2700 kg m$^{-3}$). The divergence of the fluvial/slope-wash
sediment flux can, within the valley network, be approximated by the derivative of the
volumetric unit sediment flux in the along-valley direction, $q_s$, with respect to the distance
downstream along flow lines, $x$. Adopting this approximation and summing the colluvial and
fluvial/slope-wash components, the total erosion rate at any point on the landscape is thus given
by
$$E = \frac{1}{\varepsilon_0} \frac{\partial q_s}{\partial x} - D\nabla^2 z .$$   (3)
In this paper we use equation (3) to predict the equilibrium drainage density, quantified
as the mean distance from divides to valley bottoms, associated with grasslands and shrublands
at WGEW. Specifically, we substitute an empirical power-law relationship for the mean
distance, $x$, along flow lines from divides to valley bottoms versus contributing area, $A$, into
equation (3) and solve for the value of $x$ where the fluvial erosion rate exceeds the colluvial
deposition rate by an amount equal to $E$.



Next, we further parameterize the three terms in equation (3) in terms of measured
proxies for erosion rate (e.g. decadal-scale sediment fluxes) and topographic parameters. The
erosion rate, $E$, is constrained using the ratio of the volumetric sediment flux, $Q_s$, reported by
Nearing et al. (2007) and contributing area, $A$:
$$E = \frac{1}{\varepsilon_0} \frac{Q_s}{A}. \tag{4}$$
The fluvial erosion term can be written in terms of $Q_s$ using
$$\frac{\partial q_s}{\partial x} = \frac{\partial}{\partial x}\left(\frac{Q_s}{w}\right). \tag{5}$$
where $w$ is the valley-bottom width. We approximate the mean rate of colluvial deposition at
valley heads by
$$D\nabla^2 z\Big|_{\text{heads}} \approx \frac{3DS_h}{w}, \tag{6}$$
where $S_h$ is the mean slope gradient of toe slopes as they intersect the valley bottom at valley
heads (Fig. 5). Equation (6) derives from the mass balance of a square segment of the valley
bottom in the vicinity of the valley head (Fig. 5). According to the assumption that soil-mantled
hillslopes evolve diffusively, valley bottoms in the vicinity of the valley head receive a unit
sediment flux equal to $DS_h$ by colluvial transport processes from each of three adjacent hillslope
segments. In the cross-sectional profile, the difference in sediment flux across the profile is equal
to $DS_h$ and that difference occurs over a distance of $w$. As such, the colluvial deposition rate,
computed in a manner that is independent of DEM or pixel resolution, includes the valley width
in the denominator (Pelletier, 2010a). Similarly, the divergence in the along-valley direction is
approximately $DS_h/w$. The factor of 2 does not appear in the along-valley derivative because
colluvial sediment flux enters the valley bottom segment only from the upslope direction.



Colluvial sediment flux leaving the valley head is assumed to be negligible because the slope of
the valley bottom is typically much smaller than that of the hillslope entering it from upslope.

Next, we introduce three power-law relationships that relate volumetric sediment flux to

contributing area, channel width to contributing area, and contributing area to distance along
flow lines from topographic divides. Sediment flux is a power-law function of contributing area
at WGEW (Section 3.1):
$$\frac{1}{\varepsilon_0} Q_s = k_{Qs} A^p .$$     (7)
The coefficient $k_{Qs}$ in equation (7) is a sediment transport efficiency parameter that is a function
of runoff, sediment texture, vegetation cover, and potentially other factors. It takes on different
values in the shrubland and the grassland (i.e., $k_{Qss}$ and $k_{Qsg}$), reflecting the fact that sediment
yield is a function of vegetation cover. Its value in shrublands, along with that of $p$, is obtained
by a least-squares regression of equation (7) to $Q_s$ values for the five shrubland drainage basins
of Lucky Hills reported by Nearing et al. (2007). Its value for the grasslands is constrained by
assuming that the same value of $p$ applies to both shrublands and grasslands, and using the single
$Q_s$ data point available for grasslands (i.e., for watershed 112) to constrain $k_{Qs}$ using equation (7).

Miller (1995) measured 222 cross-sectional channel profiles in the field at WGEW and

used those data to calibrate a power-law relationship between channel width and contributing
area:
$$w = k_w A^l$$     (8)
where $k_w = 0.023$ m$^{0.32}$ and $l = 0.34$. Contributing area has a power-law relationship with the
mean distance, $x$, from divides along flow lines:
$$A = k_A x^c .$$     (9)





In section 3 we report the values of $k_{Qss}$ and $k_{Qsg}$, $p$, $k_w$, $l$, $k_A$, and $c$ for WGEW. Substituting
equations (4)-(9) into equation (3) yields
$$k_{Qs} k_A^{p-1} x^{c(p-1)} = \frac{k_{Qs}}{k_w} k_A^{p-l} c(p-l) x^{c(p-l)-1} - \frac{3DS_h}{k_w k_A^l x^{cl}} . \qquad (10)$$

Equation (10) is applied using vegetation-specific values for $k_Q$ and $S_h$ to solve for the mean
distance from divides to valley heads in shrublands and grasslands, i.e., $x_s$ and $x_g$, respectively.
That is, $x_s$ is obtained by solving
$$k_{Qss} k_A^{p-1} x_s^{c(p-1)} = \frac{k_{Qss}}{k_w} k_A^{p-l} c(p-l) x_s^{c(p-l)-1} - \frac{3DS_{hs}}{k_w k_A^l x_s^{cl}} , \qquad (11)$$

and $x_g$ is obtained by solving
$$k_{Qsg} k_A^{p-1} x_g^{c(p-1)} = \frac{k_{Qsg}}{k_w} k_A^{p-l} c(p-l) x_g^{c(p-l)-1} - \frac{3DS_{hg}}{k_w k_A^l x_g^{cl}} . \qquad (12)$$

Equation (3) is generally applicable within the fluvial network. Once the colluvial deposition rate
is approximated using equation (6) (which makes use of $S_h$, the mean gradient of the hillslope toe
slopes at valley heads), subsequent equations, including equations (10)-(12), apply only to valley
heads. Equation (10)-(12) are a mathematical representation of the conceptual model, first
proposed by Tarboton et al. (1992), that erosion at valley heads is a competition between
transport-limited fluvial erosion and colluvial deposition. That is, fluvial erosion rates must
exceed colluvial deposition rates by an amount equal to the net erosion rate on the landscape in
order to maintain an equilibrium drainage density.

**3. Results**
**3.1. Topographic analyses**



Figure 3 illustrates the results of the drainage network identification for 1.6 km x 1.6 km
examples of the landscape in the vicinity of the Lucky Hills and Kendall sites. The mean
distance along flow lines from divides to valley bottoms is 18 m in the shrubland area and 50 m
in grassland area, using the Pelletier (2013) algorithm. Figures 4A&4B illustrate that hillslopes
in the shrubland area are more finely dissected with rills and gullies than in the grassland area.
As part of this analysis we also measured the mean slope gradient of hillslopes immediately
adjacent to valley heads, i.e. $S_h$ in equation (10). We obtained $S_{hs} = 0.17$ m/m in shrublands and
$S_{hs} = 0.19$ m/m in grasslands.
Mean hillslope relief increases substantially across the shrubland-to-grassland transition
(Fig. 5). Between elevations of approximately 1320 and 1430 m a.s.l., mean relief is uniformly
low (approx. 0.5-1 m). Above elevations of approximately 1450 m, hillslope relief increases
abruptly and continues to increase with increasing elevation.
Contributing area follows a piece-wise power-law function of distance along flow lines
from topographic divides (Fig. 7), with one set of values for $k_A$ and $c$ applicable on hillslopes and
another set of values applicable within the valley network. Above contributing areas of
approximately 50 m$^2$ (or, equivalently, distances from the divide equal to approximately 15 m),
contributing area increases as the 2.5 power of distance from the divide for both grassland and
shrubland areas, i.e. $k_A = 0.3$ m$^{-0.5}$ and $c = 2.5$ in equation (9). Below contributing areas of 50 m$^2$,
$k_A = 2$ m$^{-0.75}$ and $c = 1.25$. We used $k_A = 0.3$ m$^{-0.5}$ and $c = 2.5$ when solving equations (11) and
(12), since these values are most applicable to points within the valley network (i.e. at valley
heads and points downstream). This is a self-consistent approach because the solutions to
equations (11)&(12) are larger than 15 m (Section 3.2).



Plots of mean topographic curvature (i.e. the Laplacian of $z$) versus contributing area
(Fig. 8) indicate that mean curvatures are nearly identical at small and large contributing areas
but differ substantially within the range of contributing areas from ~30 to 300 m$^2$.
Plots of mean along-channel slope versus contributing area (Fig. 9) for the shrubland and
grassland area follow typical patterns for fluvial topography, i.e. the data follow a power-law
relationship for relatively large contributing areas and deviate from power-law scaling at small
contributing areas due to the predominance of colluvial processes at hillslope scales. The
exponents of the slope-area relationships differ somewhat between the shrubland and grassland
areas, i.e. 0.15 for the shrubland and 0.18 for the grassland areas. The slope-area plots also
deviate from power-law scaling at different spatial scales. Data from shrublands maintain a
power law down to contributing areas of approximately 50 m$^2$, while data from the grassland
areas deviate at larger spatial scales corresponding to contributing areas of approximately 100
m$^2$. This finding is consistent with the higher drainage density measured in the shrubland-
dominated portions of the landscape compared to the grassland portions.
**3.2. Mathematical modeling**
In this section we use the results of Section 3.1., together with analyses of the sediment
yield reported by Nearing et al. (2007), to constrain the terms in equations (11)&(12) in order to
solve for the mean distance from divides to valley heads in shrublands and grasslands. In order to
constrain the absolute values of $k_{Qss}$ and $p$, we performed a least-squares minimization of
equation (11) to the decadal-scale sediment yields reported by Nearing et al. (2007) for
watersheds 102-106 (Lucky Hills) (Table 1). This regression yields $k_{Qss} = 2\text{x}10^{-6} \pm \text{m}^{1.56}\,\text{yr}^{-1}$ (with
a range of values including one standard error from $2\text{x}10^{-7}$ to $2\text{x}10^{-5}$), $p = 1.44 \pm 0.2$, and $R^2 =$
0.93 (Fig. 10). Assuming that the value of $p$ derived from the shrubland watersheds also applies



to the grassland watershed, the value of $k_{Q_s}$ for the grassland is estimated to be $k_{Q_{sg}} = 6 \times 10^{-8}$ m$^{1.56}$
yr$^{-1}$. For $D$ we adopt the value of $1 \times 10^{-3}$ m$^2$ yr$^{-1}$ commonly inferred from scarp degradation
studies in the southwestern U.S. (e.g., Hanks, 2000, Table 2 cites $D$ values for the Basin and
Range of the western U.S. of between $6.4 \times 10^{-4}$ and $2 \times 10^{-3}$ m$^2$ yr$^{-1}$ based on eight published
studies). The full list of model parameters and their values is provided in Table 2.

We used equations (11)&(12) to predict the mean distance along flow lines from divides

to valley bottoms in shrublands and grasslands. The predicted values are $x_s = 16$ m and $x_g = 66$ m
(Table 3). In the topographic analysis shown in Figure 3, we measured 18 m in shrublands and
60 m in grasslands. As such, the model predicts mean distances from divides to valley bottoms
within 10% of measured values.

Figure 11 plots the magnitude of the three terms for a range of possible mean distances

from divide to valley bottom. The fluvial erosion rate is ~$10^{-2}$ m yr$^{-1}$ in shrublands and ~$10^{-3}$ m
yr$^{-1}$ in grasslands, increasing with distance downstream, reflecting the nonlinear relationship
between sediment flux and drainage area (Fig. 10). The colluvial deposition rate is ~$10^{-3}$ m yr$^{-1}$
for both shrublands and grasslands and decreases modestly with increasing distance from the
divide as a result of the increase in channel width with increasing contributing area (equation
(8)). Figure 11 demonstrates that the fluvial erosion rate must be quite large (approximately two
orders of magnitude larger than the net erosion rate in this case) in order to counteract the effects
of colluvial deposition and thus maintain a valley head.

We used the mean curvature versus contributing area data plotted in Figure 8 to

reconstruct an average longitudinal profile from divides to valley bottoms in shrublands and
grasslands in order to infer the approximate relief reduction associated with the late Holocene
shift from grasslands/woodlands to shrublands in WGEW. Figure 12 illustrates the results of this





integration. Integrating the curvature versus contributing area data in Figure 8 twice leads to two
constants of integration, one of which is constrained by the requirement that the slope along flow
pathways at divides is zero and the other by using a constant reference elevation at a contributing
area of approximately 300 m$^2$. We chose $A \approx 300$ m$^2$ as the location to enforce the reference
elevation because this is the contributing area below which the mean curvature begins to deviate
significantly between the grassland and shrubland areas. The difference in elevation between the
two profiles plotted in Figure 12 provides an estimate of the minimum erosion or relief reduction
associated with the shift from grassland to shrubland in the lower elevations of WGEW. We
consider the results of Figure 12 to be a minimum estimate because systematic differences in
mean slope between the grassland and shrubland across a wide range of scales (Fig. 9) are not
reflected (or not fully reflected) in curvature or any reconstruction of the longitudinal profile
based on integrating the curvature. The results in Figure 12 suggest that divides have lowered by
a minimum of approximately 0.3 m as a result of Holocene vegetation changes. Given hillslope-
scale erosion rates of approximately ~$10^{-4}$ m yr$^{-1}$ in shrublands and the much smaller erosion rate
of ~$10^{-5}$ m yr$^{-1}$ in grasslands, it would take approximately 3 kyr following a transition from
grasslands/woodlands to shrublands to erode the landscape by an amount 0.3 m greater in
shrublands compared to grasslands/woodlands. This estimate is comparable to the age of the
grassland/woodland-to-shrubland transition in the region at an elevation of 1287 inferred from
paleovegetation studies in the region, i.e., 2-4 [14]C kyr B.P. (Holmgren, 2005).

**4. Discussion**
**4.1. Uncertainty in parameter values and their impact on the model results**



Equations (11)&(12) predict mean distances from divides to valley bottoms that are
broadly similar to measured values. Several factors limit the accuracy of the comparison between
measured and predicted values. First, we relied upon a regional value for the diffusivity $D$
because we do not have a reliable means of calibrating this value locally. Second, decadal-scale
erosion rates computed from sediment samplers may differ from long-term erosion rates.
Nearing et al. (2007) found that for the six watersheds considered here, half of the sediment yield
measured from 1995-2005 was derived from 6-10 events and that the largest events contributed
between 9 and 11% of the total yield. Thus, while transport is highly episodic in WGEW, the
most effective flood events have a sufficient number of recurrences to provide an estimate of the
yield that does not depend sensitively on the time scale. This conclusion is consistent with the
fact that erosion rates inferred from sediment sampling from 1995-2005 are similar to erosion
rates measured over the post-bomb period using [137]Cs (Nearing et al., 2005). That said, the
sediment yields reported by Nearing et al. (2007) may not include the most extreme drought
conditions or other disturbances that could cause long-term sediment yields to be larger than
those reported in Table 1.
**4.2. Further discussion of the hypothesis of a vegetation-change-driven increase in drainage**
**density in the shrublands of WGEW**
We also propose that difference in mean curvatures between shrublands and grasslands
between contributing areas of ~10 and ~300 m$^2$ partly reflects recent expansion of the drainage
network in the shrublands of WGEW. This hypothesis is consistent with the fact that the
deviation of curvature values between the two sites begins at a contributing area comparable to
the support area (i.e. the contributing area required to form a valley head) in the grasslands. As
such, we propose that the shrubland areas previously had support areas comparable to the



grassland areas and that drainage network expansion has influenced the drainage network and the
morphology of the adjacent hillslopes down to spatial scales corresponding to contributing areas
of ~10 m$^2$. The fact that curvature values are very similar between shrubland and grassland
below spatial scales ~10 m$^2$ is consistent with the hypothesis that hillslopes in the lower
elevations of WGEW have not yet fully adjusted to the increase in drainage density associated
with the grassland-to-shrubland transition.

Figure 5 demonstrates that mean hillslope relief increases substantially across the

shrubland-to-grassland transition. We propose that some of this difference in hillslope relief is a
consequence of the difference in fluvial/slope-wash erosion rates, i.e. that higher erosion rates in
the lower-elevation shrublands have caused relief reduction in the past few thousand years
relative to the higher-elevation grasslands, and that this difference in fluvial/slope-wash erosion
rates is the result of a geologically recent increase in drainage density in the shrublands of
WGEW. However, it is likely that a portion of the difference in mean hillslope relief across the
study site also reflects variable uplift rates, i.e. the fact that the uplift of any piedmont or foothill
region tends to increase towards the mountain range. Flexural-isostatic response to erosion
(which has been proposed to be an important component of late Cenozoic landscape evolution in
southern Arizona (Menges and Pearthree, 1989; Pelletier, 2010b)) of the Dragoon Mountains can
be expected to have caused eastward tilting of WGEW, i.e. higher uplift rates in the higher
elevations of WGEW compared to the lower elevations. Tilting would not explain the abrupt
increase in relief at elevations just above 1430 m a.s.l., however, since no faulting occurs in the
vicinity of this contour. Therefore, it is likely that some of the difference in hillslope-scale relief
across the shrubland-to-grassland transition at WGEW is a result of the difference in erosion
rates between the shrublands and grasslands. While we can be certain of grassland-to-shrubland



shift only during the present interglacial period, the Quaternary period has seen many interglacial
periods broadly similar in climate to the current period, hence it is likely that the lower
elevations of WGEW have seen grassland-to-shrubland conversions more than once over the past
approximately 2 Myr. Each of these episodes could have contributed to relief reduction of the
lower elevations of the study site relative to the higher elevations.
Previous studies at WGEW have emphasized the role of base-level lowering and
vegetation changes within the past 130 years on the differences in erosion rate between Lucky
Hills and Kendall (Nearing et al., 2007). However, recent paleovegetation studies have provided
a new perspective. Specifically, Holmgren's (2005) documentation of shrubland species in the
region several thousand years before present at an elevation less than 100 m lower than Lucky
Hills suggests that the lower elevations of WGEW likely shifted from a grassland/woodland to a
shrubland prior to the 1880s. While base-level lowering has steepened hillslopes and channels
close to the main-stem channel of Walnut Gulch, hillslope-scale relief and slope gradients are
clearly larger at Kendall than at Lucky Hills (Figs. 6&9), indicating that base-level lowering may
be a dominant factor only for those areas within close proximity to the main channel. The
magnitude of the differences in topography (i.e., drainage density and the magnitude of erosion
than can be inferred from the change) is difficult to fit into a period as short as 130 years. Given
erosion rates measured over the past sixty at Lucky Hills, approximately 2 cm of erosion can be
expected to have occurred over the past 130 years. Figure 12 suggests that erosion associated
with a recent increase in drainage density is likely at least ten times this value, and thus more
consistent with a vegetation change that occurred several thousand years before present.
**4.3. Implications for our understanding of the controls on drainage density**



The model of this paper contributes to our broader understanding of the controls on
drainage density and it provides a mathematical model for predicting drainage density that may
be useful in other study sites.
Previous studies have demonstrated that drainage density is controlled by relief (e.g.
Montgomery and Dietrich, 1992; Tarboton, 1992; Tucker and Bras, 1999), climate (Melton,
1957; Abrahams and Ponczynski, 1984), parent material (e.g. Ray and Fischer, 1960; Day,
1980), and time (e.g. Ruhe, 1952; Dohrenwend et al., 1987). While many studies have
demonstrated the importance of individual factors on drainage density, we lack a comprehensive
model for drainage density that integrates all of these factors. Equation (10) represents one
possible candidate for such a model in soil-mantled, transport-limited landscapes. Time is not
included in the model because it is based on an equilibrium mass-balance framework.
Nevertheless, equation (10) predicts the drainage density to which a transient landscape will
likely approach over time following a perturbation.
Relief enters the model via the erosion rate, $E$ (quantified for the case of WGEW using
multiple measurements of $Q_s/A$), and the mean toe slope gradient near valley heads, $S_h$. Climate
and vegetation cover enters the model through the parameters $D$ (which increases with increasing
soil moisture and temperature cycling around 0°C (which together drive soil creep) and
increasing vegetation cover (which drives bioturbation)) and $k_{Qs}$ (which increases with rainfall
and decreases with vegetation cover)). In addition, equation (10) explicitly includes channel
width and its scaling with contributing area, factors that, to our knowledge, have not been
included in previous mathematical models of drainage density.
Drainage density is most commonly found to be an inverse function of mean annual
precipitation or effective precipitation. This finding is consistent with the conceptual model of



this paper that vegetation cover is the predominant climate-related variable that influences
drainage density, and that vegetation cover and drainage density are inversely related. Melton
(1957), for example, documented an inverse correlation between drainage density and the
precipitation-effectiveness (P-E) index at over eighty sites in the southwestern United States
including arid (low elevation) and humid (high-elevation) climates. A similar negative
correlation between drainage density and mean annual precipitation was found by Abrahams and
Ponczynski (1984). Naively, one might expect more precipitation to result in greater channel
incision and hence less contributing area, between divides and valley heads (and hence a higher
drainage density), all else being equal. Melton (1957), however, proposed that greater aridity
results in a lower vegetation density and, hence, a reduction in the cohesive strength protecting
soils on hillslopes, thus leading to a higher drainage density. One might also expect a lower
vegetation density to increase the runoff-to-rainfall ratio and hence also increase drainage
density, but runoff intensity varied by only a factor of two across Melton's sites while drainage
density varied by nearly two orders of magnitude. Istanbulluoglu and Bras (2007) provided
theoretical support for Melton's interpretation, illustrating that a lower vegetation densities can
lead to higher drainage densities through the cohesive or anchoring effect of plant roots.
**4.4. Implications for our understanding of erosion-climate linkages**
There has been an ongoing debate in the geomorphic literature regarding the importance
of climate (not limited to but often defined as mean annual precipitation (MAP)) on erosion
rates. Given the significant correlation between MAP and erosion rates in many studies within
individual mountain ranges (e.g., Reiners et al., 2003; Bookhagen and Strecker, 2012), it is
perhaps surprising how little correlation exists between MAP and erosion rates in global
compilation/synthesis studies (e.g. von Blanckenburg, 2005; Portenga and Bierman, 2011). Even



studies that emphasize the climatic control on erosion rates note that $R^2$ values between erosion
rates and MAP are quite small (e.g., Yanites and Kesler, 2015).
Recent work on the role of vegetation, and its changes through time, can provide a basis
for understanding the relatively low correlation between erosion rates and MAP in global
compilation studies and the complex relationship between erosion rates and climate more
generally. For example, Torres Acosta et al. (2015) recently documented a negative correlation
between erosion rates and both vegetation cover and MAP in Kenya. They proposed that the
primary effect of more humid conditions is to increase vegetation cover on hillslopes, thereby
reducing erosion rates on otherwise similar slope gradients. This concept is consistent with the
classic Langbein and Schumm (1958) curve. Langbein and Schumm (1958) proposed that
sediment yields are maximized in semi-arid climates (all else being equal) because such climates
generate sufficient rainfall to detach and transport soil in overland/rill flow but insufficient
vegetation cover to protect/anchor the soil. As MAP increases in this conceptual model, more
precipitation is available to drive erosion, but this effect is more than offset by a decrease in the
susceptibility of soil to erode due to the increased anchoring effect associated with greater plant
cover/biomass. The results of this paper demonstrate further complexity in the erosion-climate
relationship, i.e., that the change in climate (and hence of vegetation cover) can be as important
or more important that its mean state. That is, erosion rates can be larger during a humid-to-arid
transition than during an arid-to-humid transition, even if the mean climatic conditions (averaged
over the transition) are equal. It is important to emphasize that the effect of vegetation on the rate
of erosion by colluvial processes may be entirely different than its effect on fluvial/slope-wash
processes. All else being equal, increased vegetation cover is likely to increase erosion rates by
colluvial processes, since more plants can be expected to drive higher rates of bioturbation (e.g.,



Osterkamp et al., 2011). As such, it is crucial to consider colluvial and fluvial/slope-wash
processes separately when considering the effects of vegetation on hillslope erosion.

**5. Conclusions**

In this study we leveraged all relevant data from a uniquely well-studied semi-arid

watershed to test the hypothesis that late Holocene vegetation changes can modulate drainage
density, hillslope-scale relief, and watershed-scale erosion rates. We documented that areas
below 1430 m a.s.l. have decadal-scale erosion rates approximately ten times higher, drainage
densities approximately three times higher, and hillslope-scale relief approximately three times
lower than elevations above 1430 m. We calibrated all the terms of a mathematical landscape
evolution model and used the model to predict the equilibrium drainage density associated with
shrublands and grasslands. Model predictions for the increase in drainage density associated with
the shift from grasslands/woodlands to shrublands are broadly consistent with measured values.
Using modern erosion rates and the magnitude of relief reduction associated with the transition
from grasslands/woodlands to shrublands, we also estimated the timing of the grassland-to-
shrubland transition in the lower elevations of WGEW to be approximately 3 ka, i.e., broadly
consistent with constraints from paleovegetation studies. Our work provides a mathematical
model for predicting equilibrium drainage density in transport-limited fluvial environments that
may be applicable in other study sites.

**Data Availability**

The DEM used in this paper can be obtained upon request from the corresponding author.

All other data used in the paper are in the published literature.




**Acknowledgements**

We wish to thank Tyson Swetnam for drafting Figure 2C. J.D.P. was partially supported
by NSF award #1331408.

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





Table 1. Sediment yield data and watershed characteristics. The erosion rate calculation assumes
a soil bulk density of 1500 kg m$^{-3}$

| Watershed ID | Predominant vegetation cover type | Contributing area (ha) | Sediment yield (t ha$^{-1}$ yr$^{-1}$) | Erosion rate (mm yr$^{-1}$) |
|---|---|---|---|---|
| 102 | Shrub | 1.46 | 2.31 | 0.154 |
| 103 | Shrub | 3.68 | 5.66 | 0.377 |
| 104 | Shrub | 4.53 | 1.36 | 0.091 |
| 105 | Shrub | 0.18 | 0.75 | 0.050 |
| 106 | Shrub | 0.34 | 0.80 | 0.053 |
| 112 | Grass | 1.86 | 0.07 | 0.005 |

Table 2. List of model parameters and values.

| Symbol | Units | Description | Value |
|---|---|---|---|
| $D$ | m$^2$ yr$^{-1}$ | topographic diffusivity | 1x10$^{-3}$ |
| $S_{hs}$ | unitless | mean gradient of toe slopes (shrublands) | 0.17 |
| $S_{hg}$ | unitless | mean gradient of toe slopes (grasslands) | 0.19 |
| $l$ | unitless | exponent of width-area relationship | 0.34 |
| $k_w$ | m$^{0.32}$ | coefficient of width-area relationship | 0.023 |
| $c$ | unitless | exponent of area-distance relationship | 2.5 |
| $k_A$ | m$^{-0.5}$ | coefficient of area-distance relationship | 0.3 |
| $p$ | unitless | exponent of sediment-flux-area relationship | 1.44 |
| $k_{Qss}$ | m$^{1.56}$ yr$^{-1}$ | sediment transport coefficient (shrublands) | 2x10$^{-6}$ |
| $k_{Qsg}$ | m$^{1.56}$ yr$^{-1}$ | sediment transport coefficient (grasslands) | 6x10$^{-8}$ |

Table 3. Measured (from DEM analysis) and predicted values (from equations (11)&(12)) for the
mean distance from divides to valley bottoms in shrublands ($x_s$) and grasslands ($x_g$).

| | Measured (m) | Predicted (m) |
|---|---|---|
| $x_s$ | 18 | 16 |
| $x_g$ | 60 | 66 |




Figure 1. Maps of the bedrock geology and geomorphology of Walnut Gulch Experimental
Watershed (WGEW). (A) Bedrock geology from Osterkamp (2008). Rectangle identifies the
portion of WGEW that is the focus of this study. (B) Geomorphic map from Osterkamp (2008).
The boundary between the Dissected Whetstone Pediment and Upper Whetstone Pediment
marks a key transition in landscape morphology, soil type, and vegetation cover (Fig. 2).





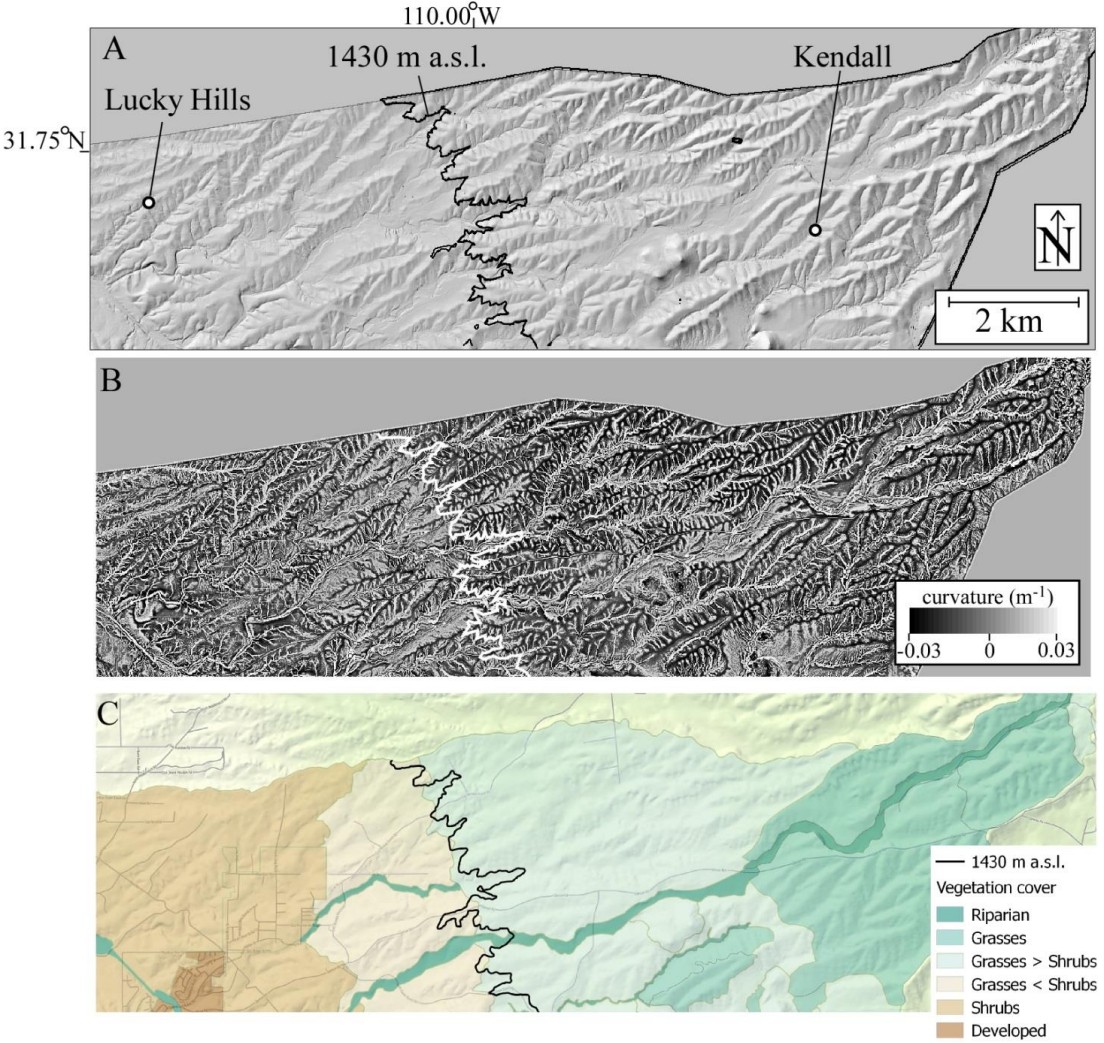

Figure 2. Relationships among landscape morphology and vegetation cover in the study area. (A)
Shaded relief image of the topography, illustrating the significant increase in hillslope-scale
relief from the western to the eastern portion of the study area. (B) Grayscale map of topographic
curvature (i.e., Laplacian), demonstrating generally lower absolute hillslope curvatures (i.e.,
more gray) in the western (shrubland) portion of the study area relative to the eastern (grassland)
portion (more black). (C) Vegetation map, after Skirvin et al. (2008), identifying the areas that
are primarily shrubland, grassland, and transitional between the two.

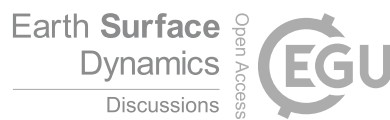



Figure 3. Drainage density is higher in shrubland areas than in grassland areas of the study site. Shaded relief images of representative 1.6 km x 1.6 km areas of (A) shrublands, including Lucky Hills watersheds 102-106 and (B) grasslands, including Kendall watershed 112. (C)&(D) Images of the drainage network identified using the Pelletier (2013) algorithm for the areas shown in (A)&(B), respectively. (E)&(F) Grayscale maps of the distance along flow lines from divides to the valley bottom. The mean of the mapped values in (E) at valley heads is 18 m while the mean of the mapped value in (F) at valley heads is 60 m.




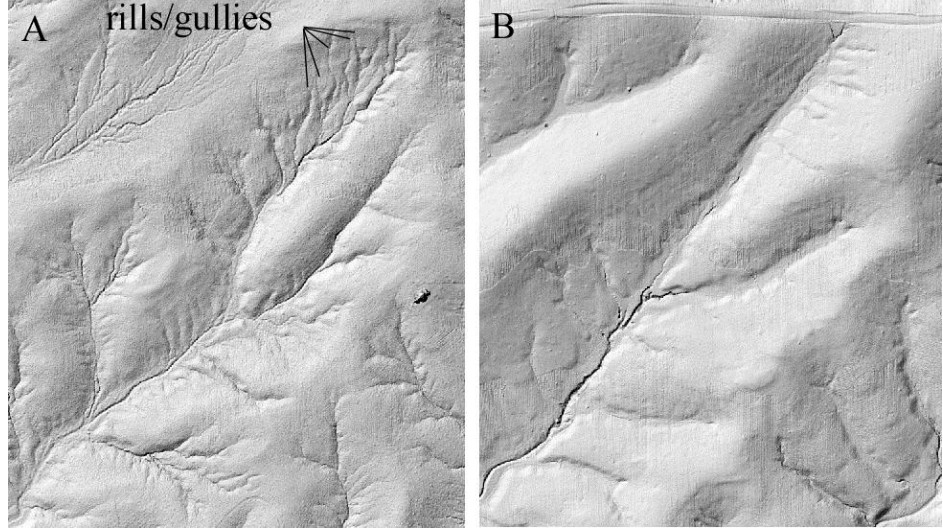

Figure 4. Detailed shaded-relief images (locations shown in Fig. 3) illustrating the presence of
parallel hillslope rills and gullies in the shrubland areas (shown in A). Grassland areas (shown in
B) have fewer such features.

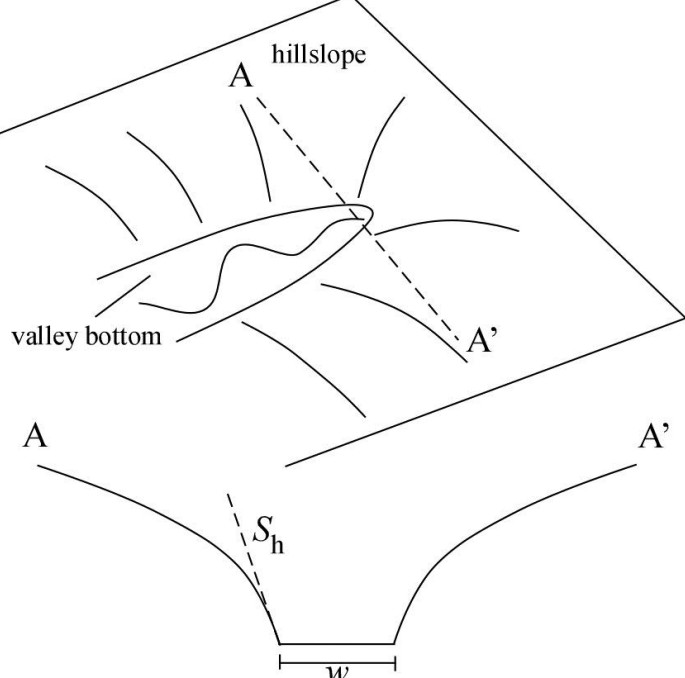

Figure 5. Schematic figure of a valley head. The profile shown along A-A' is the cross-section of
the valley head where colluvial sediment flux from hillslopes of mean gradient $S_h$ enter a
segment of width $w$. The requirement that fluvial erosion rates must be greater than colluvial




deposition rates at the valley head provides a quantitative criterion for predicting the drainage
density of landscapes.

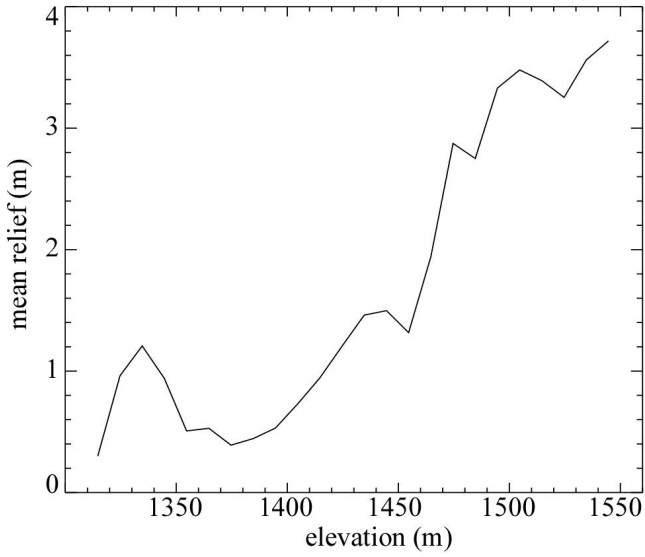

Figure 6. Plot of mean hillslope relief as a function of elevation, illustrating the marked increase
in relief above elevations of approximately 1430 m a.s.l. in the study area. Each data point
represents the mean hillslope relief in 10-m-wide elevation bins.

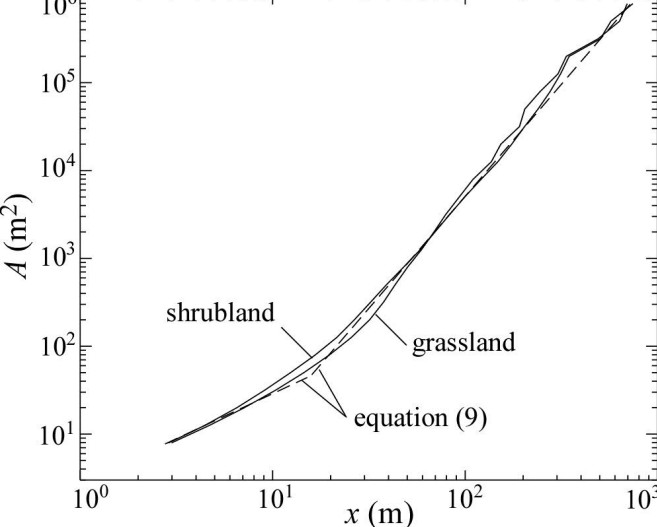

Figure 7. Plots of contributing area versus mean distance from the divide for the shrubland
(approximated as the portion of the study area below 1430 m a.s.l.) and grassland areas (above
1430 m). The dashed line plots the piece-wise power-law relationship (equation (9)) exhibited by
the data.



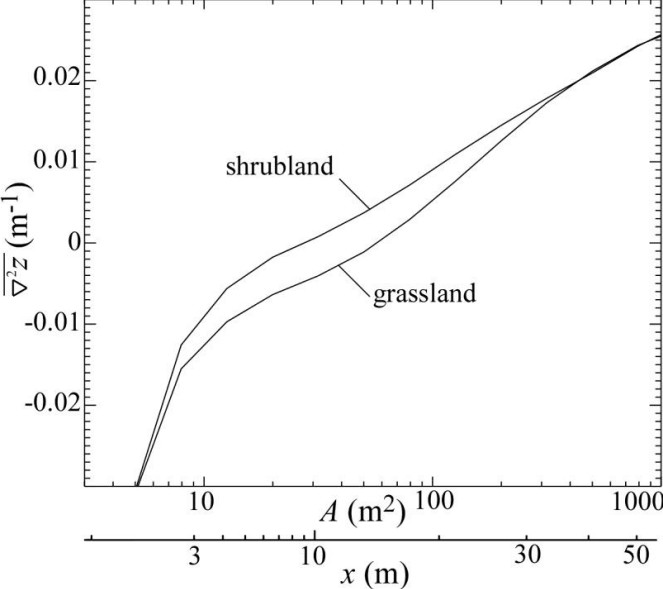

Figure 8. Plots of mean topographic curvature as a function of contributing area (distance from
divide also shown along $x$ axis using data in Fig. 7). Topographic curvatures are similar in
shrublands and grasslands at small and large contributing areas, with a minimum of 0.03 m$^{-1}$ near
divides. Within a range of contributing areas from ~10 to ~300 m$^2$ the data show significant
differences in mean curvature between shrublands and grasslands.

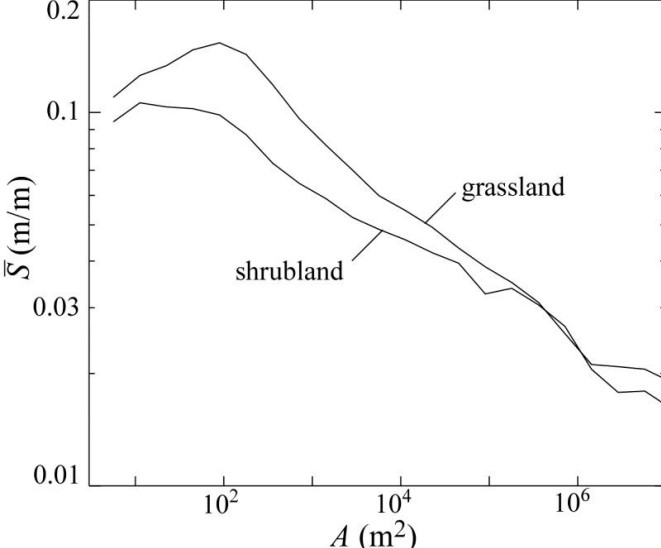

Figure 9. Plot of mean slope versus contributing area. Shrublands and grasslands both show a
power-law relationship, with deviations from power-law behavior occurring at larger spatial
scales in grasslands relative to shrublands.




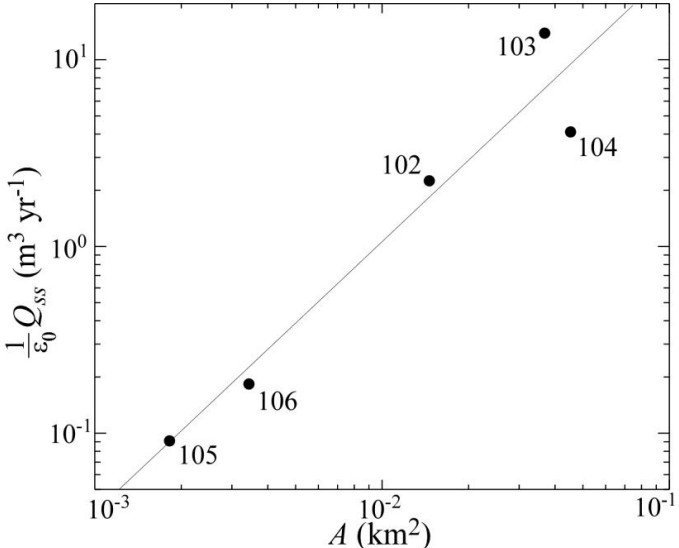

Figure 10. Plot of the decadal time-scale volumetric sediment fluxes in shrublands measured at
the five watersheds of the Lucky Hills as a function of contributing area. The straight line is the
result of a least-squares regression to the logarithms of both sides of equation (7), from which the
values of $k_{Qss}$ and $p$ were constrained.

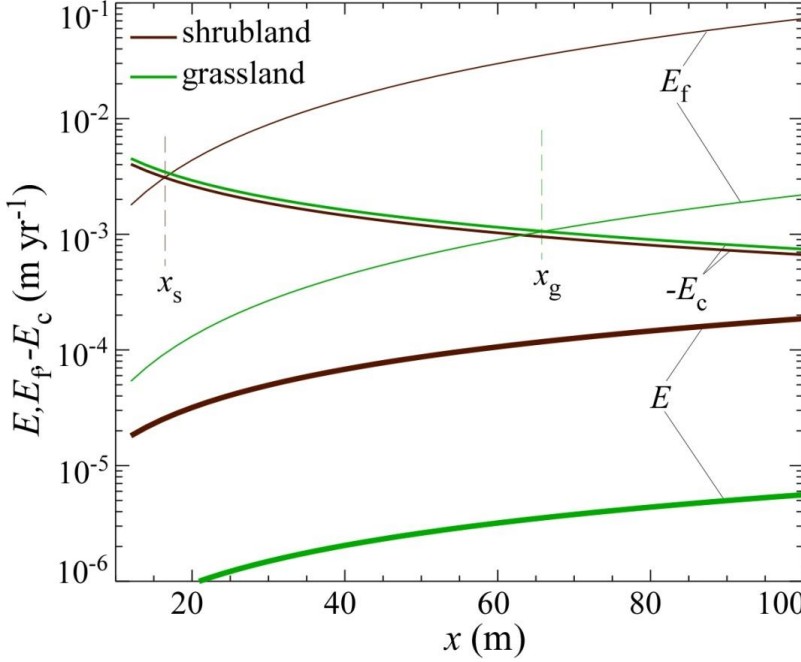

Figure 11. Plots of the total erosion rate, $E$, the fluvial/slope-wash erosion rate, $E_f$, and the
colluvial deposition rate, $-E_c$, as a function of distance along flow lines from divides, $x$. The
values of $x_s$ and $x_g$ (where $E = E_f + E_c$) are also shown.




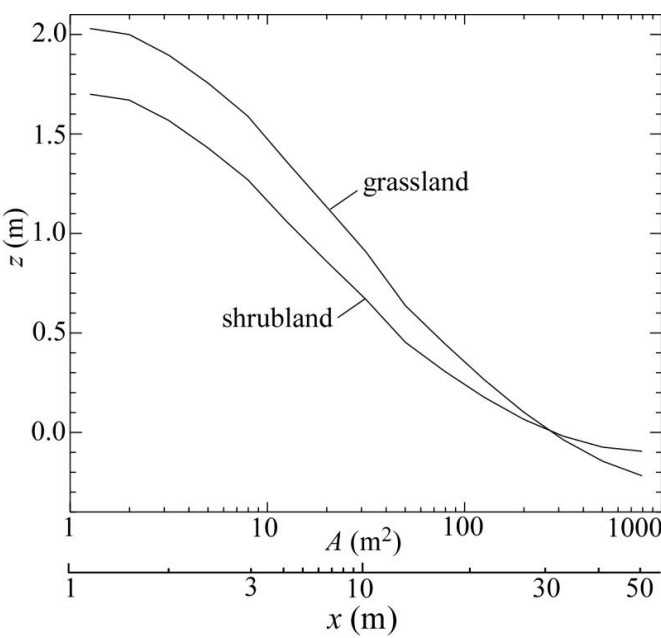

Figure 12. Plots of the mean longitudinal profile of hillslopes and valley bottoms in shrubland
and grassland areas, constructed by integrating the mean curvature data in Figure 8.