# Peer review of "The influence of Holocene vegetation changes on topography and erosion rates: A case study at Walnut Gulch Experimental Watershed, Arizona"

_Earth Surface Dynamics, 2016_

## Referee Comment (RC1) · Anonymous Referee #1 · 2 Mar 2016

In "The influence of Holocene vegetation changes on topography and erosion rates: A case study at Walnut Gulch Experimental Watershed, Arizona", the authors exploit an extensive modern data set across six catchments extending across an elevation-vegetation gradient. They combine bottom of the watershed sediment data and bare earth lidar observations and derivatives to parameterize a mathematical model that predicts equilibrium drainage density in both grassland and less-vegetated shrubland watersheds. The model results support the hypothesis that drainage density and relief differences can partly be attributed to a late Holocene vegetation change from the protective cover of grasslands and/or forests to lower-elevation shrublands with less vegetative cover and thus more erosion-susceptible bare earth.

The authors are to be commended for applying multiple data-sets from a well-studied watershed to quantify how landscapes respond to climate-driven changes in vegetation in an arid setting. However, I have one large quibble with the study and a host of lesser concerns. I am hopeful that this paper will be published post-revision, once the authors augment the current version by providing critical details and perhaps additional analysis lacking in this version of the manuscript.

The study results derive in a large part from a measured factor of 10 difference in sediment yield between 6 shrub-covered watersheds compared to one grassland watershed. My large quibble is the authors use of a modern 30-year sediment yield data set (erosion rate data) as an input parameter into a model to predict equilibrium drainage density over millennial time-scales without little discussion of the appropriateness of applying a short-term data set to evaluate how differences in erosion rates might control variations in relief, drainage density and predicted differences from divides to valley bottoms. Given that previous work (e.g. Kirchner et al., 2001) point to a discrepancy between decadal-scale measurements of stream fluxes and erosion rates over longer time scales this is a point that needs clarification and discussion in the current study While the Kirchner et al. study was in mountainous terrain, their conclusion that the episodic nature of sediment delivery may limit the use of decadal-scale sediment yield studies in evaluating long-term landscape evolution is extremely pertinent to the Walnut Gulch Experimental Watersheds. I was disappointed in the minimal discussion in the paper of potential episodic events in the watersheds not captured by the data (e.g. extreme rain events, fire) or of the potential for skew in comparing the grassland (n=1) to shrubland (n=5) watershed erosion rate data if episodic events such as fire were non-evenly represented in the watersheds. Additionally, while the mean sediment yield data reflects a 10x difference, the standard deviation for the shrub sites is quite large ($2.03 \pm 2.2$ t h$^{-1}$ yr$^{-1}$) which suggest perhaps exploring potential outcomes across a greater range of reasonable parameters. While MAP is similar between sites, the authors do not discuss differences in rainfall intensity or aspect that may also contribute to different runoff patterns and drainage density. While I am in full agreement with the stated hypothesis regarding a climate-modulated vegetation shift controlling variations in topographic attributes, additional discussion (and perhaps rejection) of alternative controls would strengthen the paper. While the author's do address some of this uncertainty in the discussion section, it seems appropriate to consider alternative scenarios earlier on.

Also I found it puzzling that while Nearing et al. 2005 concluded that the differences in sediment yields between grass and shrub sites was controlled primarily by watershed morphology this

study attributes sediment yield differences to the amount of cover. (Though admittedly there may be subsequent studies to the Nearing 2005 paper of which I am unaware).

Other general comments on the paper
It would help to clarify if any of the topographic data was smoothed and if so what was the smoothing window for deriving hillslope gradient, curvature, and the stream profiles used for extracting slope-area plots and the associated power-law fits as values as the smoothing radius choices may influence interpretation.

Also, as highlighted below in the line-specific comments, it would be helpful to report the shrubland standard error when comparing values between the grassland and shrubland sites.

Some additional justification on the use diffusive equations for colluvial sediment transport in non-soil mantled shrubland (bare earth) settings seems necessary as geomorphic transport laws do not yet exist for erosion driven by overland flow nor would I expect the hillslope erosion to be smooth but rather episodic and patchy.

Line-specific comments
L387-389      What is the SE and number of slopes evaluated for the reported slope values of 0.17 and 0.19?

L259-263 and 390-393 (Figure 6)      The relief results depicted in Figure 6 could use additional explanation regarding the 'plateaus and spikes' in the data. While overall relief does increase above ~ 1450 m, there are also rapid reversals expressed in the data.

L404-40      Why does so much of the terrain exhibit positive curvature values? This seems curious unless this corresponds to relief values and if so this is worthy of discussion. Also the deviation between grassland and shrubland curvature is correctly described in the figure caption as from ~ 10 to 300 $m^2$ rather than the 30 $m^2$ value reported in the text.

L407-417 (Figure 9)   The use of the slope-area plots seem a bit ad hoc with the exception of the identification of the colluvial region of the data sets. What part of the slope area plots were used to generate the exponents of 0.15 and 0.18 for grasslands and shrublands respectively? Is there a reason for presenting the concavity index values as they are not discussed at all in the document except to note that they somewhat different? I'm also a bit flummoxed by the use of the shrubland mean rather than plotting individual watersheds as there is no way to tell if the watersheds are similar enough to warrant combining. And finally it is common to log-bin the data to reduce noise – I am unsure how the authors generated the concavity index values. Presumably using just the fluvial portion of the plots (> 50 $m^2$ for shrublands and > 100 $m^2$ for the grasslands) and perhaps log-binned?  Also it is difficult to distinguish between the slope area plots since they both use a solid black line of the same weight. It appears as though the shrubland plot has depositional areas or knickpoints or something else leading to a non-smooth slope area relationship.

L428-431      Why rely on a diffusivity value from scarp degradation studies (which presumably are faster than hillslope diffusion) rather than the smaller diffusivity values from the

western US? This is an area where the authors could explore the parameter space using a range of values in eqns 11 and 12.

L432-435     Nice model results!

L462-468     This analysis is only valid if the decadal erosion rates apply to longer term erosion rates. The authors have not convinced me that this is true.

L523-538     Holmgren (2003) is from a region far to the south of the study area. I seem to recall that more local paleoclimate reconstructions suggest a much later transition from grasslands to shrublands. While the new perspective afforded by the more recent paleo vegetation data is understandably exciting, more context in necessary to demonstrate why this new(ish) data is applicable to a region to the north of the midden data.

L571-578     How does the reduction in vegetative influence on soil stability translate to channels – as an increase in sediment supplied to channels could potentially armor the beds? Especially given that the channels are described as transport-limited.

Section 4.4     The global compilations suggesting little correlation between MAP and erosion rates are increasingly becoming less surprising as increasing numbers of studies and conceptual models are starting to converge on periglacial temperatures and associated processes, rather than MAP, influencing erosion rates in regions mid-latitude terrain (e.g. Herman and Braun, 2008; Marshall et al., 2015; Savi et al., 2015; Schaller et al., 2002; Tucker et al., 2011). Consider modifying this section to constrain the weak linkages between MAP and erosion rates to unglaciated areas outside the influence of periglacial processes.

L601-605     Perhaps I missed it but I did not see evidence in the manuscript for erosion rates during an arid to humid transition – so while this discussion item is worth considering in general when evaluating the role of the magnitude of transient erosion states, I'm unsure if the study results can be applied to this outstanding argument.

L607-609     This statement on vegetation as an erosion agent due to bioturbation seems to directly contradict the hypothesis stated in lines 571-578. And while trees are considered both soil dilators and mechanisms for bedrock detachment and transport, I am unaware of any studies that suggest grasses or shrubs are effective mechanisms for increasing erosion.

Technical corrections
L391   Should be Figure 6 not 5.

L696-698 Holmgren 2005 should be 2003

Figure 1 Missing U/D identifiers for the fault

Figure 3 Missing identifiers of the bottom two graphics (E and F). 3E is very washed out though this may be attributed to the lower resolution.

References

Herman, F., and Braun, J., 2008, Evolution of the glacial landscape of the Southern Alps of New Zealand: Insights from a glacial erosion model: Journal of Geophysical Research, v. 113, p. F02009, doi: 10.1029/2007JF000807.

Kirchner, J.W., Finkel, R.C., Riebe, C.S., Granger, D.E., Clayton, J.L., King, J.G., and Megahan, W.F., 2001, Mountain erosion over 10 yr, 10 k.y., and 10 m.y. time scales: Geology, v. 29, p. 591–594, doi: 10.1130/0091-7613(2001)029.

Marshall, J.A., Roering, J.J., Bartlein, P.J., Gavin, D.G., Granger, D.E., Rempel, A.W., Praskievicz, S.J., and Hales, T.C., 2015, Frost for the trees: Did climate increase erosion in unglaciated landscapes during the late Pleistocene? Science Advances, v. 1, p. e1500715–e1500715, doi: 10.1126/sciadv.1500715.

Savi, S., Delunel, R., and Schlunegger, F., 2015, Efficiency of frost-cracking processes through space and time: An example from the eastern Italian Alps: Geomorphology, v. 232, p. 248–260, doi: 10.1016/j.geomorph.2015.01.009.

Schaller, M., von Blanckenburg, F., Veldkamp, A., Tebbens, L.A., Hovius, N., and Kubik, P.W., 2002, A 30 000 yr record of erosion rates from cosmogenic 10Be in Middle European river terraces: Earth and Planetary Science Letters, v. 204, p. 307–320, doi: 10.1016/S0012-821X(02)00951-2.

Tucker, G.E., McCoy, S.W., Whittaker, A.C., Roberts, G.P., Lancaster, S.T., and Phillips, R., 2011, Geomorphic significance of postglacial bedrock scarps on normal-fault footwalls: Journal of Geophysical Research: Earth Surface, v. 116, p. n/a–n/a, doi: 10.1029/2010JF001861.

---

## Author Comment (AC1) · 3 Mar 2016

We wish to thank the reviewer 1 for his/her insightful comments, which will lead to a significantly improved revision.

Q: "*My large quibble is the authors use of a modern 30-year sediment yield data set (erosion rate data) as an input parameter into a model to predict equilibrium drainage density over millennial time-scales without little discussion of the appropriateness of applying a short-term data set to evaluate how differences in erosion rates might control variations in relief, drainage density and predicted differences from divides to valley bottoms. Given that previous work (e.g. Kirchner et al., 2001) point to a discrepancy between decadal-scale measurements of stream fluxes and erosion rates over longer time scales this is a point that needs clarification and discussion in the current study While the Kirchner et al. study was in mountainous terrain, their conclusion that the episodic nature of sediment delivery may limit the use of decadal-scale sediment yield studies in evaluating long-term landscape evolution is extremely pertinent to the Walnut Gulch Experimental Watersheds. I was disappointed in the minimal discussion in the paper of potential episodic events in the watersheds not captured by the data (e.g. extreme rain events, fire) or of the potential for skew in comparing the grassland (n=1) to shrubland (n=5) watershed erosion rate data if episodic events such as fire were non-evenly represented in the watersheds. Additionally, while the mean sediment yield data reflects a 10x difference, the standard deviation for the shrub sites is quite large (2.03 ± 2.2 t h-1 yr-1) which suggest perhaps exploring potential outcomes across a greater range of reasonable parameters.*"

A: Sediment yield cannot be negative, so the measured yields should not be reported as 2.03 ± 2.2 t h$^{-1}$ yr$^{-1}$. The sediment yield is a function of drainage area, hence some of the variation in measured yields reflects a systematic increase in sediment yield with drainage area at WGEW, a dependence we accounted for. In our analysis we used the sediment yield data to quantify sediment transport coefficients that account for the drainage area dependence. For the shrubland sites the sediment transport coefficient is reported on lines 424-426 as follows: "$k_{Qss}$ = 2x10$^{-6}$ m$^{1.56}$ yr$^{-1}$ (with a range of values including one standard error from 2x10$^{-7}$ to 2x10$^{-5}$), $p$ = 1.44 ± 0.2, and $R^2$ = 0.93." The value of $k_{Qss}$ one standard error below the mean is still more than 3 times larger than the value of $k_{Qsg}$, which is 6x10$^{-8}$ m$^{1.56}$ yr$^{-1}$. We would certainly be willing to include additional potential outcomes (which would simply show a range of possible drainage densities depending on the ratio of $k_{Qss}$ to $k_{Qsg}$), but we believe that our fundamental conclusion that sediment transport coefficients are ~10x greater in shrublands relative to grasslands is not undermined by the variability in the data.

We appreciate that sediment yields can differ among time scales. However, we believe that our decadal-scale sediment yields are an appropriate estimate (and certainly the best-available estimate given the likely impossibility of [10]Be cosmogenic erosion-rate determination in WGEW due to the Pleistocene age of the bedrock) for millennial-scale yields based on three lines of argument:

1) Published analyses have shown that fluvial sediment transport in WGEW, while highly episodic, is not dominated by a very small number of extreme events. The effective discharge (i.e. the discharge above which half of the total load is transported) occurs many times within a 30-year record. In a study of sediment transport in the 1995-2005 period, for example, Nearing et al. (2007) addressed this issue as follows: "For six of the seven watersheds, between 6 and 10 events produced 50% of the total sediment yields over the 11-year period." That is, the effective discharge has a recurrence interval of between approximately 1 and 2 years. This is consistent with many other studies

demonstrating that that the effective discharge in low-gradient alluvial channels not subject to debris flows has a recurrence interval of approximately 1 to 2 years (e.g., Wolman and Miller, 1960; Andrews, 1980; Andrews and Nankervis, 1995).

References:

Andrews, E. (1980), Effective and bankfull discharge in the Yampa River Basin, Colorado and Wyoming, J. Hydrol., 46, 311-330.

Andrews, E. D., and, J. M. Nankervis (1995), Effective discharge and the design of channel maintenance flows for gravel-bed rivers, Natural and Anthropogenic Influences in Fluvial Geomorphology, AGU Monograph Series 89, Washington, D.C., 151-164.

Wolman, M. G., and, J. P. Miller (1960), Magnitude and frequency of forces in geomorphic processes, J. Geol., 68, 54-74.

2) Sediment yields calculated from 1995-2005 by Nearing et al. (2007) closely match yields measured over approximately 50 years using $^{137}$Cs (Nearing et al., 2005), as we noted on lines 205-208. The 50 year time scale includes significant droughts at WGEW. We are not suggesting that this proves that sediment yields are invariant out to time scales of millennia, but it does establish constancy of rates out to "short" time scales that are significantly longer than most studies of short-term erosion rates, including Kirchner et al. (2001).

3) Kirchner et al. (2001) showed order-of-magnitude differences in sediment yields/erosion rates between interannual and millennial-scale erosion rates from a forested landscape subject to high-severity (i.e. stand-replacing) forest fires. The recurrence interval of high-severity forest fires in the western U.S. ranges from 150 to 400 yr based on frequency distributions of even-aged stands and fire-related sedimentation studies (Meyer et al., 1995; Veblen, et al., 1994; Kipfmueller and Baker, 2000; Sibold et al., 2006; Margolis et al., 2007; 2011; Fitch and Meyer, 2015). Erosion rates commonly increase by 2 to 4 orders of magnitude for one to several years following high-severity forest fires (e.g. Wagenbrenner and Robichaud, 2014; Orem and Pelletier, 2015). These numbers strongly suggest that, *in forests similar to those studied by Kirchner et al. (2001)*, millennial-scale erosion rates may be dominated by the erosion that occurs shortly following high-severity wildfires. This suggestion is consistent with the work of Fowler (1979). In his remarkable compilation of 90 sediment yield studies in forested areas of the U.S., Fowler (1979) found that approximately 20% of the 90 studies report short-term erosion rates of <1 μm/yr, i.e. 2 orders of magnitude below typical long-term erosion rates (i.e. ~10-100 m/Myr in mid-latitude areas of low to moderate relief). Of the remaining 80% of studies he compiled (i.e. those with short-term erosion rates >1 μm/yr), Fowler (1979) demonstrated that nearly all were associated with the occurrence of wildfire and/or landsliding/debris flows. As such, it is likely that some of the episodicity in erosion rates documented by Kirchner et al. (2001) is the result of high-severity forest fires and/or episodic mass movements. Neither of these processes occurs at WGEW in the late Holocene. The modest slopes of WGEW preclude debris flows. Wildfires are of very limited size and severity in shrublands. Grassland fires in Arizona typically result in modest (if any) increase in runoff and erosion rates (e.g. Stone et al., 2003).

References:

Fitch, E.P., and Meyer, G. (2015), Temporal and spatial climatic controls on Holocene fire-related erosion and sedimentation, Jemez Mountains, New Mexico, Quat. Res., doi:10.1016/j.yqres.2015.11.008.

Fowler, J.M. (1979), The interface of forestry and agriculture as nonpoint sources of suspended sediment: A national modeling approach, Ph.D. dissertation, Iowa State Univ., Ames, Iowa, 244 p.

Kipfmueller, K.F., and Baker, W.L. (2000), A fire history of a subalpine forest in southeastern Wyoming, USA, J. Biogeogr., 27, 71-85, doi: 10.1046/j.1365-2699.2000.00364.x.

Margolis, E. Q., Swetnam, T. W., and C. D. Allen (2007), A stand-replacing fire history in upper montane forests of the southern Rocky Mountains, Can. J. Forest Res., 37(11), 2227-2241.

Margolis, E. Q., Swetnam, T. W., and C. D. Allen (2011), Historical stand-replacing fire in upper montane forests of the Madrean Sky Islands and Mogollon Plateau, southwestern USA, Fire Ecol., 7(3), 88-107.

Meyer, G.A., Wells, S.G., and Jull, A.J.T. (1995), Fire and alluvial chronology in Yellowstone National Park: Climatic and intrinsic controls on Holocene geomorphic processes, Geol. Soc. Am. Bull., 107(10), 1211-1230, doi: 10.1130/0016-7606(1995)107<1211:FAACIY>2.3.CO;2.

Orem, C., and Pelletier, J.D. (2015), Quantifying the time scale of elevated geomorphic response following wildfires using multi-temporal LiDAR data: An example from the Las Conchas fire, Jemez Mountains, New Mexico, Geomorphology, 232, 224-238, doi: 10.1016/j.geomorph.2015.01.006.

Sibold, J.S., Veblen, T.T., and Gonzalez, M.E. (2006), Spatial and temporal variation in historic fire regimes in subalpine forests across the Colorado Front Range in Rocky Mountain National Park, Colorado, USA, J. Biogegr., 32, 631-647, doi:10.1111/j.1365-2699.2005.01404.x.

Stone, J.J. et al., (2003), Post-wildfire runoff and erosion response on grassland and oak woodlands in southeastern Arizona, available at https://www.firescience.gov/projects/03-2-3-11/project/03-2-3-11_03_2_3_11_Deliverable_03.pdf

Veblen, T.T., Hadley, K.S., Nel, E.M., Kitzberger, T., Reid, M., and Villalba, R. (1994), Disturbance regime and disturbance interactions in a Rocky Mountain subalpine forest, J. Ecol., 82(1), 125-135, doi:10.2307/2261392.

Wagenbrenner, J.W., and Robichaud, P.R. (2014), Post-fire bedload sediment delivery across spatial scales in the interior western United States, Earth Surf. Process. Landf, doi:10.1002/esp.3488.

Q: *"While MAP is similar between sites, the authors do not discuss differences in rainfall intensity or aspect that may also contribute to different runoff patterns and drainage density. While I am in full agreement with the stated hypothesis regarding a climate-modulated vegetation shift controlling variations in topographic attributes, additional discussion (and perhaps rejection) of alternative controls would strengthen the paper. While the author's do address some of this uncertainty in the discussion section, it seems appropriate to consider alternative scenarios earlier on."*

A: Certainly we can include these data, which demonstrate minimal differences between the shrubland and grassland sites. Nearing et al. (2015) (referenced in our paper) demonstrated that rainfall erosivity (which includes intensity at 30 min duration) is approximately 20% lower in Lucky Hills compared to Kendall (their Figure 1b). This is somewhat larger than the difference in MAP we noted on line 145, but still insignificant when compared to the 30-fold difference in erosion rates/sediment transport coefficients. Kendall drains to the west and has roughly equal

areas of N- and S-facing hillslopes. The shrubland sites have a range of aspects and exhibit erosion rates approximately 30x higher than the grassland sites across all aspects. These points will be included in the revision.

Q: "*Also I found it puzzling that while Nearing et al. 2005 concluded that the differences in sediment yields between grass and shrub sites was controlled primarily by watershed morphology this study attributes sediment yield differences to the amount of cover. (Though admittedly there may be subsequent studies to the Nearing 2005 paper of which I am unaware).*"
A: A key point of our paper is that watershed morphology and vegetation cover are related, so it is not inconsistent to invoke both watershed morphology and vegetation cover as controls, as the two are related. Also, Nearing et al. (2005) invoked vegetation cover as a cause of erosion rate differences. In order to emphasize this fact, we took the unusual step of directly quoting Nearing et al. (2005) on this point in lines 214-220: "Nearing et al. (2005) interpreted the differences in erosion rates between Lucky Hills and Kendall to be primarily a function of vegetation cover, i.e. "hydrologic response differences as a function of vegetation differences are probably largely responsible for the differences in hillslope erosion rates between the two watersheds. If flows are more concentrated and vegetative cover is less, as on the Lucky Hills site, flow shear stresses and stream power will tend to be greater, resulting in a greater hydrologic potential for erosion. Also important is probably the higher litter cover and plant basal area cover on the grassland site that would have a direct protective effect against erosion."

Q: "*It would help to clarify if any of the topographic data was smoothed and if so what was the smoothing window for deriving hillslope gradient, curvature, and the stream profiles used for extracting slope-area plots and the associated power-law fits as values as the smoothing radius choices may influence interpretation.*"
A: As stated on line 233 an Optimal Weiner Filter was used to smooth the topography, following Pelletier (2013).

Q: "*Also, as highlighted below in the line-specific comments, it would be helpful to report the shrubland standard error when comparing values between the grassland and shrubland sites.*"
A: For the shrubland sites this value is reported on lines 424-426 as follows: "$k_{Qss} = 2x10^{-6}$ m$^{1.56}$ yr$^{-1}$ (with a range of values including one standard error from $2x10^{-7}$ to $2x10^{-5}$), $p = 1.44 \pm 0.2$, and $R^2 = 0.93$." Note that the standard error is included.

Q: "*Some additional justification on the use diffusive equations for colluvial sediment transport in non-soil mantled shrubland (bare earth) settings seems necessary as geomorphic transport laws do not yet exist for erosion driven by overland flow nor would I expect the hillslope erosion to be smooth but rather episodic and patchy.*"
A: WGEW is soil mantled almost everywhere (outcrops are extremely rare). Hence, we do not think a discussion of non-soil-mantled landscapes is relevant to our study area. It may be that the reviewer has interpreted our reference to "bare soil" on line 182 as referring to an absence of soil. Bare soil simply refers to soil with no vegetation or stone cover.

Q: "*L387-389 What is the SE and number of slopes evaluated for the reported slope values of 0.17 and 0.19?*"

A: The computed slope values are based on an analysis of all hillslope pixels in the DEM (tens of millions of pixels). It is somewhat difficult to place a precise error estimate on this value. The mean value can be estimated very precisely because we have tens of millions of pixels to average and the error of the mean decreases as sqrt(N). There is, however, also a structural error associated with the fact that what is hillslope versus what is valley bottom cannot be defined with absolute precision.

Q: "*L404-40 Why does so much of the terrain exhibit positive curvature values? This seems curious unless this corresponds to relief values and if so this is worthy of discussion. Also the deviation between grassland and shrubland curvature is correctly described in the figure caption as from ~ 10 to 300 m2 rather than the 30 m2 value reported in the text.*"
A: It may be that the community has become used to looking at numerical models of landscapes driven to topographic steady state, and that this has biased our view of the world. In such cases the vast majority of the landscape (i.e. essentially all hillslopes) will have negative curvature. In nature, however, toeslopes commonly have positive curvature and colluvial infilling often results in wide unchannelized valley bottoms with positive curvature. The numerical model results presented by Pelletier (2011) that were motivated by WGEW and do not assume topographic steady state (they were subjected to a pulse of uplift/base-level fall followed by stasis) show that much of the landscape has positive curvature. In the revision we will change the 10 m$^2$ reported value to 30 m$^2$ on line 406.

Q: "*L407-417 (Figure 9) The use of the slope-area plots seem a bit ad hoc with the exception of the identification of the colluvial region of the data sets. What part of the slope area plots were used to generate the exponents of 0.15 and 0.18 for grasslands and shrublands respectively? Is there a reason for presenting the concavity index values as they are not discussed at all in the document except to note that they somewhat different? I'm also a bit flummoxed by the use of the shrubland mean rather than plotting individual watersheds as there is no way to tell if the watersheds are similar enough to warrant combining. And finally it is common to log-bin the data to reduce noise – I am unsure how the authors generated the concavity index values. Presumably using just the fluvial portion of the plots (> 50 m2 for shrublands and > 100 m2 for the grasslands) and perhaps log-binned? Also it is difficult to distinguish between the slope area plots since they both use a solid black line of the same weight. It appears as though the shrubland plot has depositional areas or knickpoints or something else leading to a non-smooth slope area relationship.*"
A: Yes, the data were log-binned. We will take this figure and the associated text out of the revision. We just thought that analysis was useful to show as part of a complete topographic analysis of the study area, but we will remove it since the results are not essential.

Q: "*L428-431 Why rely on a diffusivity value from scarp degradation studies (which presumably are faster than hillslope diffusion) rather than the smaller diffusivity values from the western US? This is an area where the authors could explore the parameter space using a range of values in eqns 11 and 12.*"
A: Scarp degradation is modeled as hillslope diffusion by Hanks (2000) and all of the papers he cites. As such, we are unsure what the reviewer is referring to when he/she suggests a difference between diffusivity values derived from scarp degradation and those derived from hillslope diffusion. Scarp degradation is hillslope diffusion so there is no difference. The scarp studies

referenced by Hanks (2000) that led to the 1 $m^2$/kyr value are all from the western U.S. and range between approximately 0.5 and 2 $m^2$/kyr, hence they have a geometric mean of approximately 1 $m^2$/kyr.

Q: "*L523-538 Holmgren (2003) is from a region far to the south of the study area. I seem to recall that more local paleoclimate reconstructions suggest a much later transition from grasslands to shrublands. While the new perspective afforded by the more recent paleo vegetation data is understandably exciting, more context in necessary to demonstrate why this new(ish) data is applicable to a region to the north of the midden data.*"
A: The study sites of Holmgren et al. (2003) are directly east of WGEW (both are centered on 31.75°N). There is no more local paleovegetation reconstruction that exists.

Q: "*L571-578 How does the reduction in vegetative influence on soil stability translate to channels – as an increase in sediment supplied to channels could potentially armor the beds? Especially given that the channels are described as transport-limited.*"
A: A difference in bed armoring between the shrubland and grassland sites would not result from a difference in sediment supply but rather from a significant difference in the texture of the bed material. Bed sediment texture is similar in Lucky Hills and Kendall.

Q: "*Section 4.4 The global compilations suggesting little correlation between MAP and erosion rates are increasingly becoming less surprising as increasing numbers of studies and conceptual models are starting to converge on periglacial temperatures and associated processes, rather than MAP, influencing erosion rates in regions mid-latitude terrain (e.g. Herman and Braun, 2008; Marshall et al., 2015; Savi et al., 2015; Schaller et al., 2002; Tucker et al., 2011). Consider modifying this section to constrain the weak linkages between MAP and erosion rates to unglaciated areas outside the influence of periglacial processes.*"
A: We intended this discussion to focus on non-glaciated areas outside the dominant influence of periglacial processes. Sentence will be modified in revision to: "Recent work on the role of vegetation, and its changes through time, can provide a basis for understanding the relatively low correlation between erosion rates and MAP in unglaciated areas outside the dominant influence of periglacial processes and the complex relationship between erosion rates and climate in such areas more generally." That said, while periglacial processes involve temperature cycling near the freezing point of water, they also involve water, hence MAP or other measures of water availability are still relevant to the discussion of peri/paraglacial process rates.

Q: "*L601-605 Perhaps I missed it but I did not see evidence in the manuscript for erosion rates during an arid to humid transition – so while this discussion item is worth considering in general when evaluating the role of the magnitude of transient erosion states, I'm unsure if the study results can be applied to this outstanding argument.*"
A: Correct – our study provides no data for the landscape response to arid-to-humid transitions. We will simply remove the sentence "That is, erosion rates can be larger during a humid-to-arid transition than during an arid-to-humid transition, even if the mean climatic conditions (averaged over the transition) are equal" in the revision.

Q: "*L607-609 This statement on vegetation as an erosion agent due to bioturbation seems to directly contradict the hypothesis stated in lines 571-578. And while trees are considered both*

*soil dilators and mechanisms for bedrock detachment and transport, I am unaware of any studies that suggest grasses or shrubs are effective mechanisms for increasing erosion.*"

A: We do not see a contradiction here. Lines 571-578 refer to the concept that less vegetation cover leads to an increase in drainage density, all else being equal. Drainage density is set by a competition between diffusive (colluvial) and advective (fluvial) processes. Less vegetation cover leads to a decrease in colluvial transport rates and an increase in fluvial transport rates, as stated on lines 607-609, hence it increases drainage density by increasing the relative importance of advective processes relative to diffusive processes, as stated on lines 571-578.

Previous studies have shown that more vegetation cover drives higher rates of colluvial (i.e. bioturbative) transport (e.g. Hughes et al., 2009) while less vegetation cover drives higher rates of fluvial erosion (e.g. this paper and many references therein, including Abrahams et al., 1995), all else being equal. We agree there may be no studies in which an increase in grass or shrub density specifically has been shown to increase colluvial transport rates. However, there are certainly a number of studies (e.g. Hughes et al., 2009) that demonstrate that an increase in biomass generally increases sediment transport by bioturbation.

References:

Hughes, M.W., P.C. Almond, and J.J. Roering (2009), Increased sediment transport via bioturbation at the last glacial-interglacial transition, Geology, 37(10), 919-922, doi:10.1130/G30159A.1

The incorrect figure number on line 391 and the incorrect date on Holmgren et al. (2003) in the reference list will be fixed in the revised paper. We will also include labels E and F in Fig. 3. We appreciate the reviewer pointing out these errors.

---

## Referee Comment (RC3) · Anonymous Referee #2 · 25 Mar 2016

The paper is well-written and aims to advance the understanding ecosystem controls on landscape patterns. The only issue I would like to raise is in the last paragraph of this review. First I summarized the paper, practically for my understanding and kept it in the review below.

The paper investigates an interesting and relatively unique problem in the intersection between ecosystems and geomorphology at the Walnut Gulch Experimental Watershed (WGEW). The paper first introduces the paleo-ecologic change at the WGEW and resulting differences in erosion rates. In summary, areas higher than 1430 m ASL

[Figure]

have been grassland and woodlands while elevations lower than 1430 m changed from grassland and woodland to shurblands during the approximately last 2K-4K years. This led to decadal time scale erosion rates ten times higher in shrublands than grassland sites based on Nearing and coworkers' data. Drainage densities are found approximately three times higher, and relief three times lower than elevations above 1430 m where vegetation remained grassland/woodland.

The paper first uses 1m-scale DEMs to examine and show topographic differences in grassland (Kendall) and shrubland (Luck Hills) sites at the WGEW. Authors attribute the observed topographic differences to Holocene vegetation change.

The paper investigates the emergence of the above mentioned patterns using an equilibrium analytical model that predicts drainage density given different erosion rates representing shrub and grass conditions based on the equilibrium model of Tarboton et al. (1992). Decadal sediment flux data from Nearing and coworkers was used to characterize sediment flux in grass/woodland and shrubland watersheds. Channel initiation, and thus drainage density is related to the distance from the hilltop to a location where erosion by fluvial processes exceed diffusive infilling. Watershed topographic data was also used to relate contributing area to distance to outlet for different vegetation types, which is then used in the model. Model predictions were found consistent with observed topographic patters in shrub and grass/woodland vegetated sites.

The contribution of this paper is that it provides a methodology to incorporate watershed-scale sediment flux measurements to the equilibrium model that predicts drainage density. As I understand it, this model can be used where there is differential erosion measurements. Here the application site appears to be locations where vegetation may be responsible for the differences in sediment yields.. Therefore the study considers ecosystem processes implicitly and is not designed to improve ecogeomorphic modeling theory per-se.

The data analysis section of the paper is great and clearly shows associations between vegetation and morphology. The main issue I have is that an equilibrium model does not seem to be the right tool to test the hypothesis that "late Holocene vegetation changes can modulate drainage density, hillslope-scale relief, and watershed-scale erosion rates.." According to this paper all these changes might have happened approximately in the last 3K years. This hypothesis requires a transient model which would examine if the observed decadal erosion rates, when used with a conservation law, can modify landscapes such that an initially identical topography erodes faster, reducing relief and developing increased drainages that can be recognized on an evolved topography, which would have similar patterns to observed landscapes. I wonder why the authors specifically used an equilibrium model instead of using some of the existing models the lead author uses in his research. Exploring the hypothesis posed in the paper using a transient 2D or 3D model, perhaps in addition to the analytical model presented in the paper, would make this paper a lot stronger and more complete. I strongly recommend the authors to consider revising their papers with this in mind. I also list a few papers below which are relevant to the subject studied in this paper and present similar vegetation controls on landscape morphology, which the authors may want to use for comparison with their findings.

Yetemen, O., E. Istanbulluoglu, J. H. Flores-Cervantes2, E. R. Vivoni, and R. L. Bras (2015), Ecohydrologic role of solar radiation on landscape evolution, Water Resour. Res., 51. Yetemen O. , E. Istanbulluoglu, and E.R. Vivoni (2010). The implications of geology, soils, and vegetation on landscape morphology: Inferences from semi-arid basins with complex vegetation patterns in Central New Mexico, USA. Geomorphology, 116, 246–263. Istanbulluoglu E., O. Yetemen, E.R. Vivoni, H.A. Gutierrez-Jurado, and R.L. Bras (2008). Eco-geomorphic implications of hillslope aspect: Inferences from analysis of landscape morphology in central New Mexico. Geophysical Research Letters, 35, L14403, doi:10.1029/2008GL034477. Flores-Cervantes, J.H. E. Istanbulluoglu, E.R. Vivoni, and R.L. Bras (2014). A geomorphic perspective on terrain-modulated organization of vegetation productivity: Analysis in two semiarid grassland ecosystems in Southwestern United States. Ecohydrol., 7: 242–257. doi:

10.1002/eco.1333

**ESurfD**

Interactive
comment

---

## Author Comment (AC2) · 19 Apr 2016

We wish to thank the reviewers for their insightful comments, which have led to a significantly improved paper. We previously responded to reviewer 1's comments. Here we take the opportunity to add some additional comments and make clearer how the manuscript has or has not been revised to address his/her concerns. In addition, we have included our responses to reviewer 2. Finally, a tracked-changes version of the revised manuscript is provided.

**Reviewer 1:**

Q: "*My large quibble is the authors use of a modern 30-year sediment yield data set (erosion rate data) as an input parameter into a model to predict equilibrium drainage density over millennial time-scales without little discussion of the appropriateness of applying a short-term data set to evaluate how differences in erosion rates might control variations in relief, drainage density and predicted differences from divides to valley bottoms. Given that previous work (e.g. Kirchner et al., 2001) point to a discrepancy between decadal-scale measurements of stream fluxes and erosion rates over longer time scales this is a point that needs clarification and discussion in the current study While the Kirchner et al. study was in mountainous terrain, their conclusion that the episodic nature of sediment delivery may limit the use of decadal-scale sediment yield studies in evaluating long-term landscape evolution is extremely pertinent to the Walnut Gulch Experimental Watersheds. I was disappointed in the minimal discussion in the paper of potential episodic events in the watersheds not captured by the data (e.g. extreme rain events, fire) or of the potential for skew in comparing the grassland (n=1) to shrubland (n=5) watershed erosion rate data if episodic events such as fire were non-evenly represented in the watersheds. Additionally, while the mean sediment yield data reflects a 10x difference, the standard deviation for the shrub sites is quite large (2.03 ± 2.2 t h-1 yr-1) which suggest perhaps exploring potential outcomes across a greater range of reasonable parameters.*"

A:      Sediment yield cannot be negative, so the measured yields should not be reported as $2.03 \pm 2.2$ t h$^{-1}$ yr$^{-1}$. The sediment yield is a function of drainage area, hence some of the variation in measured yields reflects a systematic increase in sediment yield with drainage area at WGEW, a dependence we accounted for. In our analysis we used the sediment yield data to quantify sediment transport coefficients that account for the drainage area dependence. For the shrubland sites the sediment transport coefficient is reported on lines 424-426 as follows: "$k_{Qss} = 2x10^{-6}$ m$^{1.56}$ yr$^{-1}$ (with a range of values including one standard error from $2x10^{-7}$ to $2x10^{-5}$), $p = 1.44 \pm 0.2$, and $R^2 = 0.93$." The value of $k_{Qss}$ one standard error below the mean is still more than 3 times larger than the value of $k_{Qsg}$, which is $6x10^{-8}$ m$^{1.56}$ yr$^{-1}$. We would certainly be willing to include additional potential outcomes (which would simply show a range of possible drainage densities depending on the ratio of $k_{Qss}$ to $k_{Qsg}$), but we believe that our fundamental conclusion that sediment transport coefficients are ~10x greater in shrublands relative to grasslands is not undermined by the variability in the data.

        We appreciate that sediment yields can differ among time scales. However, we believe that our decadal-scale sediment yields are an appropriate estimate (and certainly the best-available estimate given the likely impossibility of [10]Be cosmogenic erosion-rate determination in WGEW due to the Pleistocene age of the bedrock) for millennial-scale yields based on three lines of argument:

   1) Published analyses have shown that fluvial sediment transport in WGEW, while highly episodic, is not dominated by a very small number of extreme events. The effective discharge (i.e. the discharge above which half of the total load is transported) occurs many times within a 30-year record. In a study of sediment transport in the 1995-2005

period, for example, Nearing et al. (2007) addressed this issue as follows: "For six of the seven watersheds, between 6 and 10 events produced 50% of the total sediment yields over the 11-year period." That is, the effective discharge has a recurrence interval of between approximately 1 and 2 years. This is consistent with many other studies demonstrating that that the effective discharge in low-gradient alluvial channels not subject to debris flows has a recurrence interval of approximately 1 to 2 years (e.g., Wolman and Miller, 1960; Andrews, 1980; Andrews and Nankervis, 1995).
References:
Andrews, E. (1980), Effective and bankfull discharge in the Yampa River Basin, Colorado and Wyoming, J. Hydrol., 46, 311-330.
Andrews, E. D., and, J. M. Nankervis (1995), Effective discharge and the design of channel maintenance flows for gravel-bed rivers, Natural and Anthropogenic Influences in Fluvial Geomorphology, AGU Monograph Series 89, Washington, D.C., 151-164.
Wolman, M. G., and, J. P. Miller (1960), Magnitude and frequency of forces in geomorphic processes, J. Geol., 68, 54-74.

2) Sediment yields calculated from 1995-2005 by Nearing et al. (2007) closely match yields measured over approximately 50 years using [137]Cs (Nearing et al., 2005), as we noted on lines 205-208. The 50 year time scale includes significant droughts at WGEW. We are not suggesting that this proves that sediment yields are invariant out to time scales of millennia, but it does establish constancy of rates out to "short" time scales that are significantly longer than most studies of short-term erosion rates, including Kirchner et al. (2001).

3) Kirchner et al. (2001) showed order-of-magnitude differences in sediment yields/erosion rates between interannual and millennial-scale erosion rates from a forested landscape subject to high-severity (i.e. stand-replacing) forest fires. The recurrence interval of high-severity forest fires in the western U.S. ranges from 150 to 400 yr based on frequency distributions of even-aged stands and fire-related sedimentation studies (Meyer et al., 1995; Veblen, et al., 1994; Kipfmueller and Baker, 2000; Sibold et al., 2006; Margolis et al., 2007; 2011; Fitch and Meyer, 2015). Erosion rates commonly increase by 2 to 4 orders of magnitude for one to several years following high-severity forest fires (e.g. Wagenbrenner and Robichaud, 2014; Orem and Pelletier, 2015). These numbers strongly suggest that, *in forests similar to those studied by Kirchner et al. (2001)*, millennial-scale erosion rates may be dominated by the erosion that occurs shortly following high-severity wildfires. This suggestion is consistent with the work of Fowler (1979). In his remarkable compilation of 90 sediment yield studies in forested areas of the U.S., Fowler (1979) found that approximately 20% of the 90 studies report short-term erosion rates of <1 μm/yr, i.e. 2 orders of magnitude below typical long-term erosion rates (i.e. ~10-100 m/Myr in mid-latitude areas of low to moderate relief). Of the remaining 80% of studies he compiled (i.e. those with short-term erosion rates >1 μm/yr), Fowler (1979) demonstrated that nearly all were associated with the occurrence of wildfire and/or landsliding/debris flows. As such, it is likely that some of the episodicity in erosion rates documented by Kirchner et al. (2001) is the result of high-severity forest fires and/or episodic mass movements. Neither of these processes occurs at WGEW in the late Holocene. The modest slopes of WGEW preclude debris flows. Wildfires are of very limited size and severity in shrublands. Grassland fires in Arizona typically result in modest (if any) increase in runoff and erosion rates (e.g. Stone et al., 2003).

Q: "*Also, as highlighted below in the line-specific comments, it would be helpful to report the shrubland standard error when comparing values between the grassland and shrubland sites.*"
A: For the shrubland sites this value is reported on lines 424-426 as follows: "$k_{Qss} = 2x10^{-6}$ m$^{1.56}$ yr$^{-1}$ (with a range of values including one standard error from $2x10^{-7}$ to $2x10^{-5}$), $p = 1.44 \pm 0.2$, and $R^2 = 0.93$." Note that the standard error is included. No change made on this point.

Q: "*Some additional justification on the use diffusive equations for colluvial sediment transport in non-soil mantled shrubland (bare earth) settings seems necessary as geomorphic transport laws do not yet exist for erosion driven by overland flow nor would I expect the hillslope erosion to be smooth but rather episodic and patchy.*"
A: WGEW is soil mantled almost everywhere (outcrops are extremely rare). Hence, we do not think a discussion of non-soil-mantled landscapes is relevant to our study area. It may be that the reviewer has interpreted our reference to "bare soil" on line 182 as referring to an absence of soil. Bare soil simply refers to soil with no vegetation or stone cover. No change made on this point.

Q: "*L387-389 What is the SE and number of slopes evaluated for the reported slope values of 0.17 and 0.19?*"
A: The computed slope values are based on an analysis of all hillslope pixels in the DEM (tens of millions of pixels). It is somewhat difficult to place a precise error estimate on this value. The mean value can be estimated very precisely because we have tens of millions of pixels to average and the error of the mean decreases as sqrt(N). There is, however, also a structural error associated with the fact that what is hillslope versus what is valley bottom cannot be defined with absolute precision. No change made on this point.

Q: "*L404-40 Why does so much of the terrain exhibit positive curvature values? This seems curious unless this corresponds to relief values and if so this is worthy of discussion. Also the deviation between grassland and shrubland curvature is correctly described in the figure caption as from ~ 10 to 300 m2 rather than the 30 m2 value reported in the text.*"

A: It may be that the community has become used to looking at numerical models of landscapes driven to topographic steady state, and that this has biased our view of the world. In such cases the vast majority of the landscape (i.e. essentially all hillslopes) will have negative curvature. In nature, however, toeslopes commonly have positive curvature and colluvial infilling often results in wide unchannelized valley bottoms with positive curvature. The numerical model results presented by Pelletier (2011) that were motivated by WGEW and do not assume topographic steady state (they were subjected to a pulse of uplift/base-level fall followed by stasis) show that much of the landscape has positive curvature. In the revision we changed the 30 $m^2$ reported value to 10 $m^2$ on line 406.

Q: "*L407-417 (Figure 9) The use of the slope-area plots seem a bit ad hoc with the exception of the identification of the colluvial region of the data sets. What part of the slope area plots were used to generate the exponents of 0.15 and 0.18 for grasslands and shrublands respectively? Is there a reason for presenting the concavity index values as they are not discussed at all in the document except to note that they somewhat different? I'm also a bit flummoxed by the use of the shrubland mean rather than plotting individual watersheds as there is no way to tell if the watersheds are similar enough to warrant combining. And finally it is common to log-bin the data to reduce noise – I am unsure how the authors generated the concavity index values. Presumably using just the fluvial portion of the plots (> 50 m2 for shrublands and > 100 m2 for the grasslands) and perhaps log-binned? Also it is difficult to distinguish between the slope area plots since they both use a solid black line of the same weight. It appears as though the shrubland plot has depositional areas or knickpoints or something else leading to a non-smooth slope area relationship.*"*
A: Yes, the data were log-binned. We will take this figure and the associated text out of the revision. We just thought that this analysis was useful to show as part of a complete topographic analysis of the study area, but we have removed it since the results are not essential. Figure and text removed.

Q: "*L428-431 Why rely on a diffusivity value from scarp degradation studies (which presumably are faster than hillslope diffusion) rather than the smaller diffusivity values from the western US? This is an area where the authors could explore the parameter space using a range of values in eqns 11 and 12.*"*
A: Scarp degradation is modeled as hillslope diffusion by Hanks (2000) and all of the papers he cites. As such, we are unsure what the reviewer is referring to when he/she suggests a difference between diffusivity values derived from scarp degradation and those derived from hillslope diffusion. Scarp degradation is hillslope diffusion so there is no difference. The scarp studies referenced by Hanks (2000) that led to the 1 $m^2$/kyr value are all from the western U.S. and range between approximately 0.5 and 2 $m^2$/kyr, hence they have a geometric mean of approximately 1 $m^2$/kyr. No change made on this point.

Q: "*L523-538 Holmgren (2003) is from a region far to the south of the study area. I seem to recall that more local paleoclimate reconstructions suggest a much later transition from grasslands to shrublands. While the new perspective afforded by the more recent paleo vegetation data is understandably exciting, more context in necessary to demonstrate why this new(ish) data is applicable to a region to the north of the midden data.*"*

A: The study sites of Holmgren et al. (2003) are directly east of WGEW (both are centered on 31.75°N). There is no more local paleovegetation reconstruction that exists. No change made on this point.

Q: "*L571-578 How does the reduction in vegetative influence on soil stability translate to channels – as an increase in sediment supplied to channels could potentially armor the beds? Especially given that the channels are described as transport-limited.*"
A: A difference in bed armoring between the shrubland and grassland sites would not result from a difference in sediment supply but rather from a significant difference in the texture of the bed material. Bed sediment texture is similar in Lucky Hills and Kendall. No change made on this point.

Q: "*Section 4.4 The global compilations suggesting little correlation between MAP and erosion rates are increasingly becoming less surprising as increasing numbers of studies and conceptual models are starting to converge on periglacial temperatures and associated processes, rather than MAP, influencing erosion rates in regions mid-latitude terrain (e.g. Herman and Braun, 2008; Marshall et al., 2015; Savi et al., 2015; Schaller et al., 2002; Tucker et al., 2011). Consider modifying this section to constrain the weak linkages between MAP and erosion rates to unglaciated areas outside the influence of periglacial processes.*"
A: We intended this discussion to focus on non-glaciated areas outside the dominant influence of periglacial processes. Sentence will be modified in revision to: "Recent work on the role of vegetation, and its changes through time, can provide a basis for understanding the relatively low correlation between erosion rates and MAP in unglaciated areas outside the dominant influence of periglacial processes and the complex relationship between erosion rates and climate in such areas more generally." That said, while periglacial processes involve temperature cycling near the freezing point of water, they also involve water, hence MAP or other measures of water availability are still relevant to the discussion of peri/paraglacial process rates. Sentence modified as stated.

Q: "*L601-605 Perhaps I missed it but I did not see evidence in the manuscript for erosion rates during an arid to humid transition – so while this discussion item is worth considering in general when evaluating the role of the magnitude of transient erosion states, I'm unsure if the study results can be applied to this outstanding argument.*"
A: Correct – our study provides no data for the landscape response to arid-to-humid transitions. We will simply remove the sentence "That is, erosion rates can be larger during a humid-to-arid transition than during an arid-to-humid transition, even if the mean climatic conditions (averaged over the transition) are equal" in the revision. Sentence removed.

Q: "*L607-609 This statement on vegetation as an erosion agent due to bioturbation seems to directly contradict the hypothesis stated in lines 571-578. And while trees are considered both soil dilators and mechanisms for bedrock detachment and transport, I am unaware of any studies that suggest grasses or shrubs are effective mechanisms for increasing erosion.*"
A: We do not see a contradiction here. Lines 571-578 refer to the concept that less vegetation cover leads to an increase in drainage density, all else being equal. Drainage density is set by a competition between diffusive (colluvial) and advective (fluvial) processes. Less vegetation cover leads to a decrease in colluvial transport rates and an increase in fluvial transport rates, as

stated on lines 607-609, hence it increases drainage density by increasing the relative importance of advective processes relative to diffusive processes, as stated on lines 571-578.

Previous studies have shown that more vegetation cover drives higher rates of colluvial (i.e. bioturbative) transport (e.g. Hughes et al., 2009) while less vegetation cover drives higher rates of fluvial erosion (e.g. this paper and many references therein, including Abrahams et al., 1995), all else being equal. We agree there may be no studies in which an increase in grass or shrub density specifically has been shown to increase colluvial transport rates. However, there are certainly a number of studies (e.g. Hughes et al., 2009) that demonstrate that an increase in biomass generally increases sediment transport by bioturbation.

**Reviewer 2:**

Q: "*The only issue I would like to raise is in the last paragraph of this review….The data analysis section of the paper is great and clearly shows associations between vegetation and morphology. The main issue I have is that an equilibrium model does not seem to be the right tool to test the hypothesis that "late Holocene vegetation changes can modulate drainage density, hillslope-scale relief, and watershed-scale erosion rates." According to this paper all these changes might have happened approximately in the last 3K years. This hypothesis requires a transient model which would examine if the observed decadal erosion rates, when used with a conservation law, can modify landscapes such that an initially identical topography erodes faster, reducing relief and developing increased drainages that can be recognized on an evolved topography, which would have similar patterns to observed landscapes. I wonder why the authors specifically used an equilibrium model instead of using some of the existing models the lead author uses in his research. Exploring the hypothesis posed in the paper using a transient 2D or 3D model, perhaps in addition to the analytical model presented in the paper, would make this paper a lot stronger and more complete. I strongly recommend the authors to consider revising their papers with this in mind.*"

A: To address this issue we have explicitly added a transient component to the analysis. The added text is as follows:

In the Methods section:

"To estimate the time required for low-order channels to grow headward in response to hillslope vegetation changes, we modeled the channels as a diffusive system (e.g., Begin, 1988). In diffusive systems, the time scale, $t_r$, required for a response over a length scale, $L$, can be estimated using

$$t_r \sim \frac{L^2}{D} \qquad (13)$$

where $D$ is a diffusivity. The value of $D$ in the diffusive model for alluvial channels is given by the ratio of the unit volumetric sediment flux to the along-channel slope:

$$D = \frac{Q_s}{wS}.$$ (14)"

In the Results section:

"       In order to test the hypothesis that 2-4 kyr is sufficient time for drainage density to have fully responded to the recent grassland/woodland-to-shrubland transition, we used equations (13) and (14) to estimate the response time of low-order channels. We used $L = 50$ m to represent the approximate distance that the valley heads migrated upslope in response to late Holocene vegetation changes. We used data associated with the outlets of watersheds 105 and 106 as representative of the low-order channels in the shrubland-dominated portions of WGEW. Using equation (14), the $D$ value for these channels is approximately $3 \times 10^3$ $m^2$/kyr based on the ~0.1 $m^3$/yr sediment flux of the 105 and 106 watersheds (Fig. 10), a width of approximately 1 m, and slope of approximately 0.03. Substituting these values for $L$ and $D$ into equation (13) yields an estimate for $t_r$ equal to 0.83 kyr, or ≈1 kyr in keeping with the approximate nature of this calculation. This time scale is significantly shorter than the age of the transition (2-4 kyr), suggesting that sufficient time has occurred for drainage density to have fully responded to the changes in vegetation cover in the lower elevations of WGEW."

Also:

"The fact that curvature values are very similar between shrubland and grassland below spatial scales ~10 m2 is consistent with the hypothesis that hillslopes in the lower elevations of WGEW have not yet fully adjusted to the increase in drainage density associated with the grassland-to-shrubland transition. However, as the analysis of equations (13) and (14) reveal, the drainage density itself (which is the focus of this paper) has likely had sufficient time to fully adjust to the vegetation changes."

       The reviewer suggested that we use more sophisticated landscape evolution models to investigate whether 3 kyr is sufficient time for the system to fully respond to hillslope vegetation changes at this study site. This is a very reasonable suggestion. However, we think the simpler approach we have included adequately addresses the issue he/she raised. The first author appreciates that the reviewer acknowledges that he is capable of performing more sophisticated landscape evolution modeling. He choose not to include such models in this paper because 1) he thought the models presented were adequate to address the problem, 2) he wants to take the time to carefully apply such models to WGEW as part of a future study that includes additional data (e.g. soil texture on hillslopes and channels), and 3) he is currently revamping his transport-limited landscape evolution models with more sophisticated solution techniques for a number of new applications and release to the community.

Q: "*I also list a few papers below which are relevant to the subject studied in this paper and present similar vegetation controls on landscape morphology, which the authors may want to use for comparison with their findings.*"

A: We wish to thank the reviewer for these suggestions. We decided to add the Yetemen et al. (2010) reference as this one seemed most relevant. While the others are also relevant, they appear to be more focused on slope aspect controls on vegetation and slope form, which is of great interest to us but is less directly relevant to our study (which does not deal with slope aspect). Sentence added:

[revised manuscript text omitted]

---

## Author Response (AR2)

We wish to thank the AE for his kind words and thoughtful suggestions for improvement. Below is a point-by-point explanation of how we have modified the paper to address his concerns. If we have not responded adequately to any issue we would be happy to revise again.

Q: "*p.6 Line 14: Refer to the figure here. Also the "63" designation is not in the figure so this could be slightly confusing. Perhaps mention it in the figure caption.*"
A: 63 removed. We don't reference Fig. 9 here because the axis labels of the figure refer to variables in equations presented later in the paper. As such, we think this figure is best referenced in the Results section, after the reader has seen the equations and can understand how the data are used to calibrate the model parameters.

Q: "*p.6 Lines 15-22: The citations for the soil data, vegetation types, etc. are missing in this paragraph, which is odd since they are included in the following paragraph. Add the appropriate citations.*"
A: Citations added.

Q: "*p. 8, Line 21: "Multiple-flow-direction algorithm" is vague since this could refer to an entire family of flow algorithms (this could refer to any algorithm that routes flow to more than one pixel, so could include, for instance, the d-inf algorithm). Add a citation so readers know which one it is.*"
A: We used Freeman (1991). Reference added.

Q: "*p. 8, Line 22: State how were the flow lines were calculated.*"
A: Steepest-descent directions. Clarification added.

Q: "*p. 8, Line 23: This is a bit confusing since it states an averaging routine was used for "all pixels draining into the channels" but previously it was stated that the hillslope length was calculated on the basis of flow lines emanating from topographic divides. If every pixel is used, rather than those emanating from divides, the hillslope length will be underestimated significantly so it is important to clarify this point.*"
A: Rephrased: "We mapped the longest distance upslope from every hillslope pixel along steepest-descent flow lines. We then averaged this flow distance value for all pixels adjacent to valley heads." We think this is clearer but would be happy to add to or change the text if the AE can think of a better wording.

Q: "*p. 9 line 20- p 10 line 2: Culling was a pioneer and should be cited here, but he didn't show that "sediment transport by colluvial processes leads to a diffusion equation for topography if slopes are uniformly soil mantled and topographic gradients are modest". He proposed a linear flux law because it was convenient. Culling himself says that the linear flux law is "the fundamental assumption…analogous to the assumption made in the theory of heat conduction". As far as I know, there isn't much evidence for a linear flux law beyond McKean's 1993 geology paper (based on 2 data points and a line passing through the origin) and the rainsplash papers of Furbish et al 2007 and Dunne et al 2010, both in JGR. There are a number of papers that suggest that topography is consistent with a nonlinear flux law (Roering's 1999 and 2008 papers, papers from Martin Hurst and Stuart Grieve from my own group) which approximates to an erosion statement like equation (1) for low topographic gradients. So while I am perfectly*"

*happy with equation (1) and think it is justified in this setting, the text, as it is currently written, gives insufficient justification and should be modified.*"

A: The AE is correct – this was thoughtless writing. We have changed this to: "Sediment transport by colluvial processes increases approximately linearly with slope based on short-term monitoring studies (Kirkby, 1967) and cosmogenic radionuclide analysis (McKean et al., 1993) if slopes are uniformly soil-mantled and topographic gradients are modest. Linear slope-dependent transport, combined with conservation of mass, leads to a diffusion equation for topography (Culling, 1960)." We think the use of Kirkby is appropriate here even though the AE did not suggest it. Kirkby did not emphasize a slope dependence in his 1967 paper (because the scatter of the data was large) but in his book (Carson and Kirby, 1972) he concluded that the data from his 1967 paper "suggested" a linear dependence on slope.

Q: "*Line 11, line 14: I like Figure 5 but it doesn't show the square approximation of sediment flux that is used to arrive at equation (6). I think this would be clearer if, in addition to what is already there, a small figure showing the gridded abstraction of channel head geometry is used to illustrate equation (6). Basically this is in response to looking at Figure 5 and thinking "what 3 adjacent hillslopes?" and then interpreting the 3 adjacent hillslopes as the two side slopes and the one upslope pixel if you used a gridded approximation to Figure 5.*"

A: The first author is not a very good artist. He tried to draw a valley head in a computational grid framework and found that it could be more confusing to the reader. However, he has updated Figure 5 to show an along-channel in addition to a cross-section profile in order to make clear what the 3 adjacent slopes are.

Q: "*Page 12, Line 1: Refer to the actual slopes here.*"

A: Added: "Colluvial sediment flux leaving the valley head is assumed to be negligible because the slope of the valley bottom (approx. 5%) tends to be much smaller than that of the hillslopes entering it from upslope (approx. 15-20%) (data presented in Section 3.1) "

Q: "*Page 12, Line 5: I suggest adding a very brief note here that explains you will demonstrate this base on Nearing et al.'s field data.*"

A: Added: "Equation (7) will be calibrated in section 3.1 based on data from Nearing et al. (2007)."

Q: "*Page 14, Line 5: How many of these measurements were made? I ask because I wonder how noisy this data is. Also, the slope of low order channels isn't really relevant since the main purpose of presenting channel slope is to support the assumption that there is no colluvial transport out of the cell at the channel head.*"

A: The mean value was computed based on every valley head in the study site (totaling thousands of valley heads). We have included the standard deviations: "We obtained $S_{hs} = 0.17 \pm 0.04$ m/m in shrublands and $S_{hs} = 0.19 \pm 0.05$ m/m in grasslands (uncertainty is the standard deviation)." Also, we modified the wording from low-order valleys to first-order valleys (which is what we used for the calculation).

Q: "*Page 15: Lines 1-2: I very much enjoyed the way a range of careful field measurements is balanced in this paper to support the mathematical model. However, there is a component of the model that raises some questions. That is that the p value is greater than 1. This implies that*

*erosion is a function of drainage area (according to equation (4)). The erosion goes as nearly the square root of the area. So if you have the junction between two small basins of equal size, the erosion rate below the junction is ~sqrt(2) larger than the erosion rate just upstream of the junction. The erosion rate, according to this equation, suggests that as you move downstream erosion rates increase and you end up with something that looks very much like a wave of incision moving through the landscape. I do not think that the authors believe there is a wave of incision in the landscape. What do the authors think is behind this trend? What does a linear fit through figure 9 look like? I am not going to hold the paper up on this point but perhaps the authors would like to comment on the apparent dissonance between the model fit suggesting erosion is increasing downstream (with the consequence that slope angles will need to be transient) and the equilibrium assumption in equation (11) and (12), which rely on slope angles remaining constant.*"

A: If $p$ were less than or equal to 1, fluvial valleys would not form on Earth (Smith and Bretherton, 1972). The erosion rate must increase with increasing $A$ (or, equivalently, sediment flux must increase faster than proportionally with $A$) in order for the positive feedback responsible for valley formation to occur, i.e., localization of drainage into incipient depressions increases erosion rate, which in turn leads to more localized drainage, balanced only by colluvial deposition. Moreover, we don't see any reason why WGEW should be in an equilibrium condition of constant relief over any time scale.

That said, we see the AE's point: if $p > 1$, an inverse relationship between slope and area is required for the steady-state topography that many people assume exists for landscapes. Yet, we do not have an explicit slope term in our sediment flux relationship. We did attempt to calibrate our $p$ value by regressing sediment yield on both area and slope. However, the results were very similar to those using $A$ only since slope does not have a strong relationship to contributing area in WGEW for low-order valleys (as the discussion paper showed, the mean slope-area exponent for WGEW is -0.18, while typical values are -0.5 to -0.4). We don't know for sure why slope is such a weak function of area in WGEW, but one possible reason is that the landscape has undergone tilting (as we proposed in the Discussion section) rather than the usual localized base-level drop that we, as a community, commonly assume.

In the revision we have added the along-channel slope data to Table 1 and the following text: "Bed-load sediment flux is a function of both contributing area and along-channel slope. However, along-channel slopes (Table 1) vary only weakly with contributing area at the monitoring sites, i.e., along-channel slope varies by only a factor of 2 as contributing area varies by more than a factor of 20. As such, including slope does not significantly modify or improve the regression of measured data against its controlling factors (see Section 3.1). Hence, we did not include it explicitly in equation (7)." If the AE would like to see this expanded into a larger discussion point, we can do that. However, we think this is sufficient because we are not invoking steady state topography over any time scale nor is the slope-area relationship a critical input or output of our model.

Q: "*Page 15, lines 3-5: A single value of D is used, on the basis that there is no way to otherwise measure it. But curvature on ridgetops reflects D. According to figure 8 the ridgetop curvature of both sites is the same, yet there are large differences in erosion rates, suggesting that D is quite different in the two sites. The fact that the shrub landscape has the same ridgetop curvature but a much higher erosion rate than the grassland suggests that the D value is higher in the shrub landscape. This is consistent with the findings of Dunne et al (2010-JGR-ES) who showed*

*that decreasing cover can increase sediment transport rates by rain splash by orders of magnitude. The authors later mention that this assumption leads to uncertainty in the results but I feel a few more sentences on the issue would be useful, if only to refer the reader to the discussion in section 4.2 asserting that there has not been enough time to modify the ridgetops.*"

A: Curvature on ridgetops reflects the ratio of $E$ to $D$ over time scales of $>\sim10^5$ yr (estimated using the diffusion time scale $t_r \sim \lambda^2/D$ with $\lambda \sim 10$ m and $D \sim 1$ m$^2$/kyr). We have no data on erosion rates at WGEW over such long time scales, and there is no evidence that erosion rates differ between the shrubland and grassland sites over this time scale (which is a function primarily of glacial climates very different from Holocene time scales). We agree that a reduction in vegetation cover can increase the component of $D$ driven by rainsplash, but this result depends sensitively on soil texture. Of the fraction of WGEW without canopy cover, gravel and cobbles cover more than 50% of the surface. Rainsplash has a limited effect on such large particles.

Q: "*Page 18, line 21-25: Earlier the authors state that curvature values on ridgetops are not affected because the erosion signal has not had time to propagate to the ridgetops. But if this is true then increased erosion in the channels and downslope would increase relief. The fact that relief is lower in the shrubs suggests that D could have increased in the shrubland. I suggest adding a sentence or two on this point.*"

A: We are not suggesting that there has been no lowering of divides (Fig. 11 in fact proposes that the divides have lowered in the shrublands relative to the grasslands). Rather, we don't think it likely that the divides have fully adjusted their curvatures to current vegetation conditions, as the time scale for diffusive adjustment at ridgetops is $\sim10^5$ yr, assuming that curvature is computed over a length scale $\sim10$ m (or, $\sim10^4$ yr if computed over a length scale of $\sim 3$ m).

Q: "*Page 20, line 12-14: This passage needs citations. I am not aware of a paper that shows increasing vegetation cover monotonically increases D values but the Dunne et al 2010 paper suggests transport rates increase with decreasing cover. The Gabet et al. (2003) paper would suggest a positive relationship between biomass and D, but these were the results from various mathematical models and not data.*"

A: Our apologies. The first author had Hanks (2000) in his mind when I wrote this sentence, but neglected to include the reference. We have added: "Hanks (2000) compiled data on $D$ values estimated from morphological analysis of dated shorelines from the Negev in Israel, the semi-arid southwestern U.S., and sub-humid to humid areas in California and Michigan. The available data suggest that $D$ values increase from dry to wet climates and/or from areas of low to high vegetation cover: $D$ values are $\sim0.1$-$0.3$ m$^2$/kyr in arid areas, $\sim1$ m$^2$/kyr in semi-arid areas, and $\sim10$ m$^2$/kyr in sub-humid and humid areas.

Q: "*Figure 8: This plot needs some information on uncertainties since hoe else is the reader to know if these differences are real or just noise?*"

A: We have modified the figure to include the standard error of the mean values for each bin. The text added in the caption is: "The error bars represent the standard error of the mean for each bin." Please note that they do not represent the standard deviation of the data, which is much larger. We think the standard error of the mean is what is needed here, because the goal is to compare the mean values of curvature as a function of (i.e., using averages of binned data for) contributing area. Also, we note that it would be very unusual if the two curves sat precisely on

top of one another at large and small contributing areas but deviated from each other smoothly within a range of intermediate contributing areas due simply to noise.

[revised manuscript text omitted]